# SphericalDreamer: Generating Navigable Immersive 3D Worlds with Panorama Fusion

**Antoine Schnepf** [1 2]   **Karim Kassab** [1]   **Flavian Vasile** [1]   **Andrew Comport** [2]

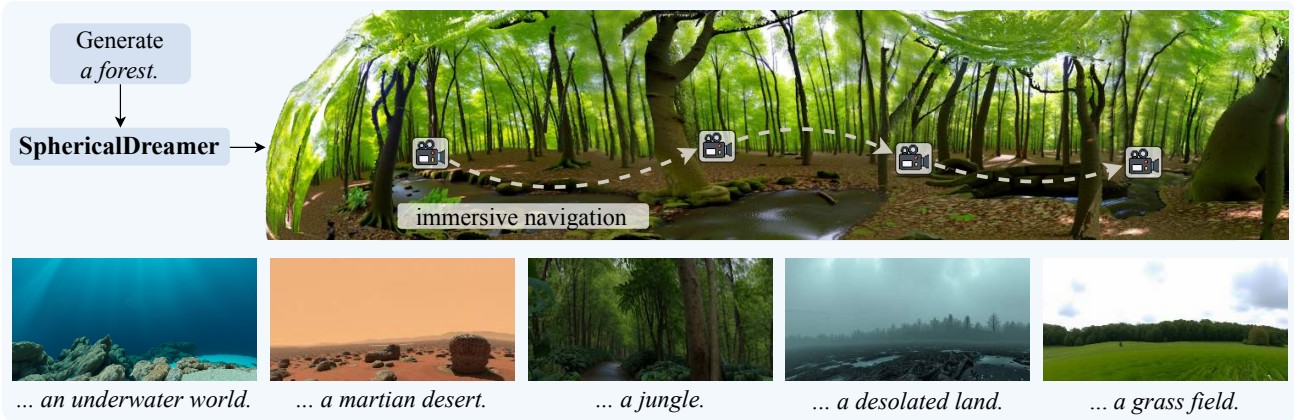

*Figure 1.* **SphericalDreamer.** Our method generates diverse 3D worlds from textual prompts, enabling immersive navigation over long distances. The bottom row illustrates views encountered during the exploration of generated environments (see Table 4 for full prompts). Results are best appreciated as videos; see our project page at https://sphericaldreamer.github.io/.

## Abstract

The generation of immersive and navigable 3D environments is increasingly prevalent with the growing adoption of virtual reality and 3D content. However, recent methods face a fundamental limitation: they cannot produce 3D worlds that simultaneously (i) are navigable over long-range spatial extents and (ii) cover the complete omnidirectional field of view ($360°$ horizontally and $180°$ vertically). To address this challenge, we introduce SphericalDreamer, a method for generating fully immersive and long-range 3D outdoor environments from textual prompts. Our approach is built on the generation of multiple panoramic images, which are subsequently lifted into 3D and fused together while maintaining visual and geometric consistency. SphericalDreamer produces highly detailed, fully immersive 3D environments, while substantially improving scale and navigability compared to prior approaches.

## 1. Introduction

The rapid development of spatial computing, including virtual and mixed reality, has created strong demand for high-quality, navigable 3D environments that enable immersive user experiences. While the manual creation of such environments remains costly, recent progress in 2D generation, depth estimation, and 3D generation has substantially lowered the cost barriers to 3D content creation, making it more broadly accessible. Nevertheless, existing methods face a fundamental limitation: they cannot generate 3D environments that are simultaneously (i) fully immersive with complete omnidirectional coverage ($360°$ horizontally and $180°$ vertically) and (ii) navigable over long distances. Specifically, prior work can be divided into two paradigms that satisfy these requirements only partially: *panorama-based* methods, which provide high immersivity but limited spatial extent, and *iterative completion* methods, which can generate long-range environments but sacrifice immersion.

**Panorama-based** approaches generate 3D environments by synthesizing high-resolution equirectangular images (i.e. panoramas) and lifting them into 3D representations (Yang et al., 2025; Zhou et al., 2025; Paliwal et al., 2025; Schwarz et al., 2025). Because such panoramas encode the complete omnidirectional light field observed from a single spatial location, these methods are particularly effective at producing fully immersive environments. However, in such

[1]Criteo AI Lab, Paris, France [2]Université Côte d'Azur, CNRS, I3S, France. Correspondence to: Antoine Schnepf .

*Proceedings of the $43^{rd}$ International Conference on Machine Learning*, Seoul, South Korea. PMLR 306, 2026. Copyright 2026 by the author(s).

approaches, camera translation is inherently limited to a small neighborhood around the panorama nodal point; larger translations introduce distortions or cause collisions with the synthesized geometry. As a result, *panorama-based* methods excel at immersion but fundamentally lack long-range navigability.

**Iterative completion** approaches iteratively expand a 3D scene by rendering novel views, completing missing regions via image inpainting, and then back-projecting the completed observations into the 3D representation (Chung et al., 2025; Yu et al., 2025; Fridman et al., 2023; Yu et al., 2024). This allows for the 3D world to be explorable by the observer. However, a fundamental constraint of this paradigm is that, during generation, the camera cannot advance in directions that have already been observed: since those regions are already completed, new content cannot be added. Therefore, covering all viewing directions at a given location leaves no space for new content, effectively closing the scene and preventing long-range navigation. To prevent such cases, these methods typically expand the environment along a backward camera trajectory, enabling long-range scene navigation. However, this design choice inherently limits viewpoint coverage (e.g. the camera cannot look behind), thus preventing immersiveness. As a result, *iterative completion* methods can generate long-range environments, but only at the expense of immersion.

To address the aforementioned challenges, we introduce SphericalDreamer, a framework that is inspired from both paradigms to produce 3D scenes that are simultaneously navigable and immersive (Figure 1). We focus on outdoor and natural environments, for which spherical imagery is particularly well-suited. Our core idea is to combine multiple *panorama-based* environments into a single, unified 3D world that is significantly larger than what can be achieved with a single panorama. To do so, we first generate multiple layered depth panoramas (LDPs) (Shih et al., 2020) from a text prompt, with separate foreground and background layers to resolve occlusions, and lift each LDP into 3D to form the building blocks of the 3D world. We also call those building blocks *spheres*, as their shapes resemble that of a sphere. To connect spheres together, we must first create connection regions. To this end, we open each sphere on its left, right, or both sides, enabling connectivity accordingly. Specifically, pairs of consecutive spheres are opened along their facing regions and arranged to form a capsule-shaped point cloud with an empty central region. This empty space allows us to generate a seamless and coherent transition using a pipeline inspired by *iterative completion* approaches. As such, a camera is placed at the capsule center to render an equirectangular view. Then, empty image regions are completed using inpainting. The depth of the synthesized content is then estimated and integrated into the existing point cloud via Harmonic Blending, a novel energy-based

Table 1. Comparison of 3D world generation methods. **Immersive** indicates support for full omnidirectional view coverage (180° vertically and 360° horizontally). **Navigable** indicates the ability to generate long-range scenes that allow extended spatial traversal.

| Method | Immersive | Navigable |
|---|---|---|
| *Iterative completion approaches* | | |
| LucidDreamer | ✗ | ✗ |
| WonderWorld | ✗ | ✗ |
| SceneScape | ✗ | ✓ |
| WonderJourney | ✗ | ✓ |
| *Panorama-based approaches* | | |
| LayerPano3D | ✓ | ✗ |
| HoloDreamer | ✓ | ✗ |
| PanoDreamer | ✓ | ✗ |
| **SphericalDreamer (Ours)** | ✓ | ✓ |

method that ensures smooth geometric transitions with the surrounding geometry. Once transition blocks are generated for each pair of consecutive spheres, the full 3D world is obtained by assembling all building and transition blocks.

We evaluate our method on a diverse set of generated outdoor and natural 3D environments using both qualitative visualizations and quantitative metrics, where it consistently outperforms existing baselines in terms of scale and immersivity, with state-of-the-art visual fidelity. Our open-source code is available from our project page at `https://sphericaldreamer.github.io/`.

## 2. Related Work

In this section, we review recent progress in generating immersive 3D environments. We first discuss neural 3D representation methods, which ensure geometric consistency but remain limited in scale and realism. We then examine image-to-3D lifting approaches that leverage powerful 2D generative models to achieve higher visual fidelity.

### 2.1. 3D generation with neural representation.

Neural implicit representations (e.g. NeRF, Mildenhall et al. (2020)), provide continuous, high-quality 3D representations of geometry and appearance. Coupled with curated 3D datasets (e.g. Objaverse, Deitke et al. (2023)) and generative frameworks like GANs (Goodfellow et al., 2014) and diffusion models (Ho et al., 2020), they have substantially advanced 3D generative models (Schwarz et al., 2020; Schnepf et al., 2023; Wang et al., 2023b), especially for object-centric generation. Nonetheless, extensions to scene-level generation remain challenging. Approaches such as GAUDI (Bautista et al., 2022) demonstrate the feasibility of neural-field-based scene generation but remain limited by scarce and low-diversity 3D scene datasets, resulting in domain-specific outputs. To mitigate data scarcity, per-

instance optimization methods guide neural 3D representations using large text-image models (Poole et al., 2023). Nonetheless, these methods remain primarily suited for object-level generation, struggling to scale to scene-level generation.

Overall, neural-representation-based methods face fundamental challenges in generating large, immersive 3D environments due to data scarcity, optimization costs, and the complexity of directly modeling full scenes in 3D.

### 2.2. 3D world generation with image-to-3D lifting.

To overcome the limitations of the aforementioned methods, another branch of the literature constructs 3D scenes from synthesized images by combining text-to-image diffusion models (e.g. Rombach et al. (2022)) with monocular depth estimation (e.g. Yang et al. (2024)). These approaches benefit from strong visual priors learned from large-scale 2D data, producing photo-realistic results. An overview of these approaches is provided in Table 1.

**Panorama-based** methods leverage panoramic image generation to produce immersive environments. Because a panorama captures the complete light field at a single spatial location, such methods naturally provide omnidirectional coverage with strong visual consistency. However, this also means that there would be occlusion artifacts under camera motion. To address this, LayerPano3D (Yang et al., 2025) generates multi-layer panoramas (e.g. foreground, background), which are then lifted into 3D. HoloDreamer (Zhou et al., 2025) handles occluded regions through inpainting along carefully designed camera trajectories. Paliwal et al. (2025); Schwarz et al. (2025) reformulate panorama synthesis as an image completion problem, where a partial view is extended into a full panorama using diffusion models, improving diffusion-based panorama generation.

Nonetheless, panorama-based methods share a fundamental limitation: since the scene is represented from a single viewpoint, camera motion is restricted to a small neighborhood around the panorama center. In contrast, our approach fuses multiple panoramas into a coherent 3D world, enabling long-range navigation while preserving immersivity.

**Iterative completion** methods expand a 3D scene by repeatedly rendering it from new viewpoints to reveal previously unseen regions. At each step, these newly exposed areas are filled via image inpainting, and the synthesized content is back-projected into the scene using monocular depth estimation. LucidDreamer (Chung et al., 2025) introduces this paradigm but is limited to local scene expansion and does not support long-range or fully immersive 3D scene generation. It is primarily suited to front-facing views. Text2Room (Höllein et al., 2023) enables immersive generation, but remains limited to indoor environments.

SceneScape (Fridman et al., 2023) and WonderJourney (Yu et al., 2024) enable long-range scene generation via backward camera trajectories. However, their generated scenes are optimized for a specific viewpoint, and deviating from it leads to strong visual artifacts.

As opposed to iterative completion methods, SphericalDreamer can produce long-range 3D world while maintaining full omnidirectional view coverage with high visual consistency, thanks to our usage of panoramic images.

## 3. Method

We introduce SphericalDreamer, a novel method that generates large-scale, immersive 3D worlds from textual descriptions. SphericalDreamer extends panorama-based 3D generation to a *multi-panorama* setting. By progressively fusing multiple text-generated panoramas into a single coherent 3D world, SphericalDreamer enables the synthesis of scenes at large spatial scales. It produces 3D worlds in which, at each spatial location, the full omnidirectional field of view is covered, supporting immersive exploration along long-range, complex camera trajectories.

Our method consists of three main stages. In **Stage I**, we start by generating the building blocks of our 3D world. We achieve this by generating Layer Depth Panoramas (LDPs) from text, and then lifting them into 3D. In **Stage II**, we fuse the spherical building blocks in a pairwise manner. For each block pair, we generate a filling block that acts as a transition region. To do so, we use 2D image inpainting combined with depth harmonic blending to first complete the missing content in the image and depth domain, and then back-project it into 3D. In **Stage III**, we assemble the full 3D world by combining the building blocks of the first stage with the filling blocks of the second stage. An overview of the method is shown in Figure 2.

**Problem formulation.** Given a textual prompt $p$ and a number $N > 1$ of panoramas to generate and fuse, our goal is to synthesize a 3D world $\mathcal{W}$ represented as a colored pointcloud:

$$\mathcal{W} = \{(\mathbf{p_k}, \mathbf{c_k})\}_{k=0}^{K-1} \,,$$

with the point positions $\mathbf{p_k} \in \mathbb{R}^3$ and corresponding point colors $\mathbf{c_k} \in [0,1]^3$. One can consider $N$ as a proxy for the desired spatial extent of the final world: increasing $N$ leads to fusing more panoramas together, leading to a longer-range generated environment.

**Initialization.** We start by defining a straight-line trajectory along a horizontal direction $\mathbf{d} \in \mathbb{R}^3$, on which we sample $N$ equidistant camera poses:

$$\{\mathbf{T}_i\}_{i=0}^{N-1} \,, \qquad \mathbf{T}_i \in SE(3) \,.$$

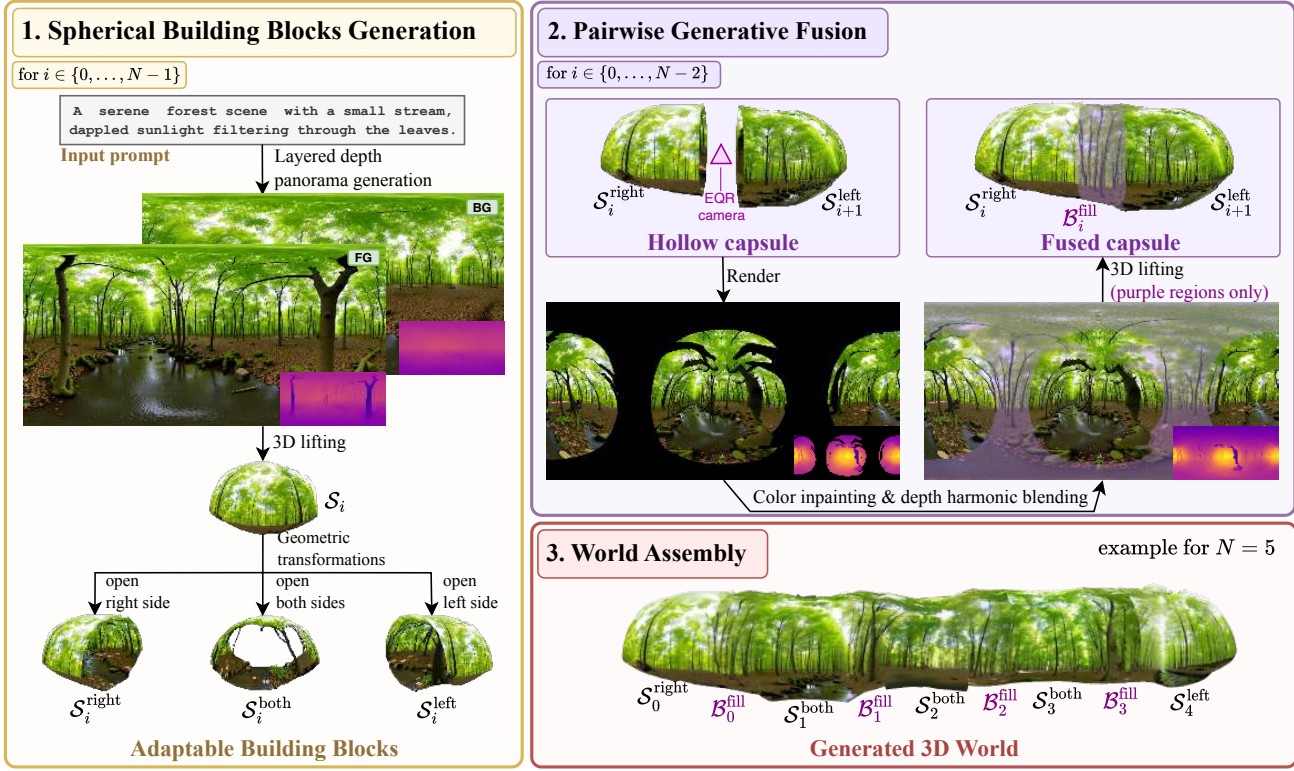

*Figure 2.* **Overview.** SphericalDreamer generates navigable immersive 3D worlds from textual prompts. In **Stage I**, a set of spherical building blocks $\{\mathcal{S}_i\}_{i=0}^{N-1}$ is generated by lifting multiple text-generated layered depth panoramas into 3D. Each block $\mathcal{S}_i$, also referred to as a sphere, can be geometrically transformed to create a connection interface on its right side, left side, or both. In **Stage II**, consecutive spheres $\mathcal{S}_i$ and $\mathcal{S}_{i+1}$ are positioned to face each other, forming a capsule-like configuration with a missing central region. An intermediate RGB–D view is rendered, its missing regions are inpainted, and the newly synthesized content is lifted into 3D to produce a filling block $\mathcal{B}_i^{\text{fill}}$, thereby completing the connection between the two spheres. In **Stage III**, all spheres and filling blocks are assembled to produce the complete 3D world.

Each successive pose is obtained by translating the previous one by a fixed displacement $\lambda \mathbf{d}$:

$$\mathbf{T}_{i+1} = \text{TRANSLATE}(\mathbf{T}_i, \lambda \mathbf{d}) , \qquad (1)$$

where $\lambda \in \mathbb{R}_+$ controls the spacing between consecutive viewpoints.

### 3.1. Spherical building blocks generation

In the first stage of our method, we start by generating a set of $N$ panoramas from the input prompt. Each panorama is then converted into a Layer Depth Panorama (LDP), and then lifted into a 3D pointcloud $\mathcal{S}_i$, centered around camera $\mathbf{T}_i$. We refer to each 3D pointcloud $\mathcal{S}_i$ as a *sphere*, as its shape resembles that of a 3D sphere. These spheres serve as the fundamental building blocks for assembling our 3D world $\mathcal{W}$. This process is detailed below.

**Panorama generation and depth estimation.** For each camera pose $\mathbf{T}_i$, we generate a corresponding equirectangular RGB panorama $I_i \in \mathbb{R}^{H \times W \times 3}$ using the prompt $p$ and

a text-conditioned panorama generation model (Yang et al., 2025) based on Flux (Black Forest Labs, 2024).

Then, for each panorama, we estimate a dense depth map $D_i \in \mathbb{R}_+^{H \times W}$ using a pretrained monocular depth estimator specifically designed for panoramic images (Rey–Area et al., 2022). Each resulting RGB–D panorama $(I_i, D_i)$ serves as the basis for constructing a spherical building block $S_i$ at camera pose $T_i$.

However, directly using these panoramas results in occlusion artifacts during navigation in 3D. This is because each panorama represents the scene from a single viewpoint and therefore contains occluded regions that become visible under camera motion, as shown in Figure 3b. To mitigate this, we convert each panorama into a layer depth panorama.

**Layered depth panorama.** Layered depth panoramas are designed to handle occlusion artifacts induced by camera motion. Given a panoramic image and its corresponding depth map $(I_i, D_i)$, we first estimate a foreground mask $M_i^{\text{fg}} \in \{0, 1\}^{H \times W}$ by jointly reasoning over image seg-

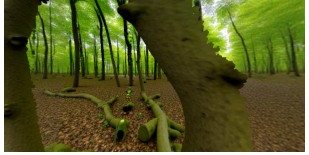
*(a)* With LDP (two layers).

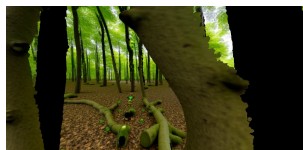
*(b)* Without LDP (one layer).

*Figure 3.* **LDP ablation.** With LDP (a), the observer can freely navigate the 3D world without encountering missing background regions. Without LDP (b), foreground occlusions remain unhandled, resulting in visible holes in the background.

mentation masks and depth discontinuities.

We start by estimating candidate segmentation masks $\{S_k\}$ using the Segment Anything Model (SAM) (Kirillov et al., 2023). Then, we select the masks representing foreground regions using a novel mask scoring technique. Each mask is scored based on two criteria: (i) its boundary alignment with corresponding edges in the depth map, and (ii) the magnitude of depth gradient normal to its boundaries. Masks with sufficiently high scores represent prominent foreground regions likely to occlude the background, and are merged into a single foreground mask $M_i^{\text{fg}}$.

Subsequently, we mask out foreground elements of $I_i$ using $M_i^{\text{fg}}$, and inpaint masked regions, producing a complete background panorama $I_i^{\text{bg}}$, similarly to Yang et al. (2025). Unlike prior approaches, we recover the depth for the inpainted background regions by constructing a background depth estimate $D_i^{\text{bg}}$. It is obtained by taking the row-wise maximum depth of the original panorama, yielding a smooth enclosing surface that captures the farthest scene extent at each elevation. This procedure yields a background RGB–D layer $(I_i^{\text{bg}}, D_i^{\text{bg}})$. Figure 6 shows the resulting background layer, while Figure 3 compares novel views rendered with and without it. More details about LDP construction are available in Appendix Section A.

Finally, each layered depth panorama $(I_i, D_i, I_i^{\text{bg}}, D_i^{\text{bg}})$, is ready to be lifted into 3D, as described in the next paragraph.

**Sphere construction via 3D lifting.** At each camera pose $\mathbf{T_i}$ we back-project the corresponding RGB–D panorama $(I_i, D_i)$ to a set of colored 3D points in the world reference frame. To do so, point positions are retrieved from $D_i$, while point colors are retrieved from $I_i$. Let $\Pi_{\mathbb{S}}^{-1}$ be the spherical back-projection operator. Then, we obtain

$$S_i = \Pi_{\mathbb{S}}^{-1}(I_i, D_i, \mathbf{T_i}).\qquad(2)$$

Similarly, background points are obtained from the background layer $(I_i^{\text{bg}}, D_i^{\text{bg}})$, yielding $S_i^{\text{bg}}$.

The final spherical point cloud is obtained by merging foreground and background points:

$$\mathcal{S}_i = S_i \cup S_i^{\text{bg}} = \{(\mathbf{p_{i,k}}, \mathbf{c_{i,k}})\}_k.\qquad(3)$$

At this point, we obtain a collection $\{\mathcal{S}_i\}_{i=0}^{N-1}$ of disjoint 3D spheres that are positioned adjacently.

**Spheres as adaptable building blocks.** To construct a single large-scale 3D world, the spheres $\{\mathcal{S}_i\}_{i=0}^{N-1}$ must be fused together, as they are currently disjoint. However, each sphere forms a closed geometric structure, preventing a direct connection with an adjacent sphere.

As such, we apply *opening* transformations to our spheres to create connection regions. Figure 2 provides a visual illustration of this process (bottom left, "Adaptable Building Blocks"). Concretely, each opening transformation first removes a subset of points from each sphere along a left or right direction, corresponding to the position of the adjacent sphere. The resulting opened spheres are then deformed onto an enclosing cylindrical surface.

With these opening operations, each sphere can act as an adaptable building block, which can be joined with another sphere from either the left, the right, or both sides.

Hence, the assembled 3D world is written as:

$$\mathcal{W}^{\text{partial}} = \mathcal{S}_0^{\text{right}} \cup \bigcup_{i=1}^{N-2} \mathcal{S}_i^{\text{both}} \cup \mathcal{S}_{N-1}^{\text{left}},\qquad(4)$$

where $\mathcal{S}_i^{\text{right}}$, $\mathcal{S}_i^{\text{left}}$, and $\mathcal{S}_i^{\text{both}}$ denote spheres opened on their right, left, or both sides respectively. It can be visualized in Figure 8 of the Appendix. Importantly, since two spheres may contain different elements at their intended connection regions, we deliberately introduce a gap between adjacent spheres by distancing our camera positions $\{\mathbf{T_i}\}_{i=0}^{N-1}$. These gaps will be completed using a generative fusion pipeline, presented in the next phase of our method.

### 3.2. Pairwise Generative Fusion

In this section, we describe how the missing regions in $\mathcal{W}^{\text{partial}}$ are completed. To fuse consecutive spheres $\mathcal{S}_i$ and $\mathcal{S}_{i+1}$, we start by rendering an intermediate viewpoint that reveals the missing regions, and subsequently inpaint these regions. The inpainted regions are then lifted into 3D while ensuring proper alignment with existing geometry, yielding a *filling* block $\mathcal{B}_i^{\text{fill}}$. We apply this operation for all $i \in \{0, \dots, N-2\}$.

To fuse consecutive spheres $\mathcal{S}_i$ and $\mathcal{S}_{i+1}$ and obtain the filling block $\mathcal{B}_i^{\text{fill}}$, we proceed in multiple steps described below.

**Hollow capsule construction.** Using the spheres states presented in the previous section, we place the two spheres $\mathcal{S}_i$ and $\mathcal{S}_{i+1}$ in a natural setup for connecting them: the left-most panorama $\mathcal{S}_i$ is opened on its right side, yielding $\mathcal{S}_i^{\text{right}}$, and the right-most panorama $\mathcal{S}_{i+1}$ is opened on its left side,

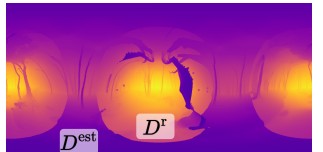 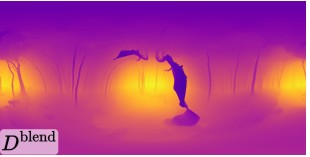

*(a)* Naive blending.    *(b)* Harmonic blending.

*Figure 4.* **Depth harmonic blending.** Blending an estimated depth map $D^{\mathrm{est}}$ into a trusted depth map $D^{\mathrm{r}}$ using (a) naive blending produces visible depth discontinuities at the blending boundary, whereas (b) harmonic blending yields a seamless, consistent depth map $D^{\mathrm{blend}}$ .

yielding $\mathcal{S}_{i+1}^{\mathrm{left}}$ . This setup orients the regions of spheres that should be connected in a facing position, forming a capsule-like shape with a missing central part.

**Intermediate rendering.**    To fill the missing central part of the capsule-like pointcloud, we first define the intermediate camera pose

$$\mathbf{T}_{i+\frac{1}{2}} = \text{TRANSLATE}\big(\mathbf{T}_i, \tfrac{1}{2}\lambda\mathbf{d}\big), \qquad (5)$$

where $\lambda\mathbf{d}$ is the known translation step. This pose corresponds to a camera located at exactly the middle of the capsule.

Subsequently, we render an EQR panorama and corresponding depth with the intermediate camera pose :

$$I_i^{\mathrm{r}}, D_i^{\mathrm{r}}, M_i^{\mathrm{r}} = \text{RENDER-EQR}\Big(\mathcal{S}_i^{\mathrm{right}} \cup \mathcal{S}_{i+1}^{\mathrm{left}}, \ \mathbf{T}_{i+\frac{1}{2}}\Big), \tag{6}$$

where $I_i^{\mathrm{r}}$ and $D_i^{\mathrm{r}}$ designates the rendered RGB–D panorama and $M_i^{\mathrm{r}} \in \{0,1\}^{H \times W}$ designates a binary visibility mask indicating pixels hit by at least one projected point. Pixels outside the valid region defined by $M_i^{\mathrm{r}}$ are not hit by any point in the pointcloud and therefore correspond to regions of missing information.

**RGB inpainting.**    We complete the missing panorama regions detected in the previous step using RGB inpainting with FluxFill (Black Forest Labs, 2024):

$$I_i^{\mathrm{ip}} = \text{INPAINT}(I_i^{\mathrm{r}}, 1 - M_i^{\mathrm{r}}, p), \qquad (7)$$

where $p$ denotes the global text prompt, and $I_i^{\mathrm{ip}}$ the inpainted image. This procedure yields a completed RGB panorama that provides a coherent appearance for the previously missing regions of the capsule.

**Depth harmonic blending.**    To lift newly generated content into 3D, we first estimate a depth map for the inpainted image using a monocular depth estimator, yielding $D_i^{\mathrm{est}}$. Nonetheless, this estimated depth differs from the reference depth $D_i^{\mathrm{r}}$ over its validity region $M_i^{\mathrm{r}}$. As illustrated in Figure 4a, naïvely replacing the unknown depth values of $D_i^{\mathrm{r}}$

with the estimated ones would introduce visible seams and geometric discontinuities in the reconstructed scene.

To enforce a smooth transition, we align estimated depth $D_i^{\mathrm{est}}$ on reference depth $D_i^{\mathrm{r}}$ using a novel *harmonic blending* pipeline, formulated as a graph-based energy minimization and inspired by classical Laplacian mesh editing and harmonic surface deformation (Sorkine et al., 2004; Yu et al., 2004). Newly synthesized points are deformed by minimizing a Laplacian smoothness energy on a k-NN graph, subject to Dirichlet constraints that enforce exact agreement with existing geometry along the blending boundary. This formulation yields a smooth displacement field that preserves local structure while ensuring geometric alignment. Technical details about harmonic blending are available in Appendix B.

We denote by $D_i^{\mathrm{blend}}$ the depth map obtained from the harmonic blending of the estimated depth $D_i^{\mathrm{est}}$ in the reference depth $D_i^{\mathrm{r}}$ :

$$D_i^{\mathrm{blend}} = \text{HARMONIC-BLEND}\big(D_i^{\mathrm{r}}, \ D_i^{\mathrm{est}}, M_i^{\mathrm{r}}\big). \tag{8}$$

Figure 4b visualizes the harmonically blended depth, demonstrating a smooth transition at the blending boundaries.

**3D lifting.**    Subsequently, $(I_i^{\mathrm{ip}}, D_i^{\mathrm{blend}})$ is lifted to 3D, yielding the filling block $\mathcal{B}_i^{\mathrm{fill}}$:

$$\mathcal{B}_i^{\mathrm{fill}} = \Pi_{\mathbb{S}}^{-1}(I_i^{\mathrm{ip}}, \ D_i^{\mathrm{blend}}, \ \mathbf{T}_{i+\frac{1}{2}}, \ 1 - M_i^{\mathrm{r}}), \qquad (9)$$

where we restrict the lifting to the regions of newly synthesized content defined by the mask $(1 - M_i^{\mathrm{r}})$. By construction, $\mathcal{B}_i^{\mathrm{fill}}$ fills the missing regions in the capsule formed by $\mathcal{S}_i^{\mathrm{right}}$ and $\mathcal{S}_{i+1}^{\mathrm{left}}$ .

### 3.3. World assembly

In this final stage, all spheres and filling blocks are assembled, yielding the final world point cloud

$$\mathcal{W} = \mathcal{W}^{\mathrm{partial}} \cup \bigcup_{i=0}^{N-2} \mathcal{B}_i^{\mathrm{fill}}, \qquad (10)$$

where we recall that $\mathcal{W}^{\mathrm{partial}}$ is the partially assembled 3D world as defined by Equation (4).

## 4. Experiments

We evaluate SphericalDreamer against recent baselines along three axes: (i) rendering quality, (ii) scene immersivity, and (iii) scene navigability. Our evaluation focuses on outdoor and natural environments, for which spherical imagery is best suited.

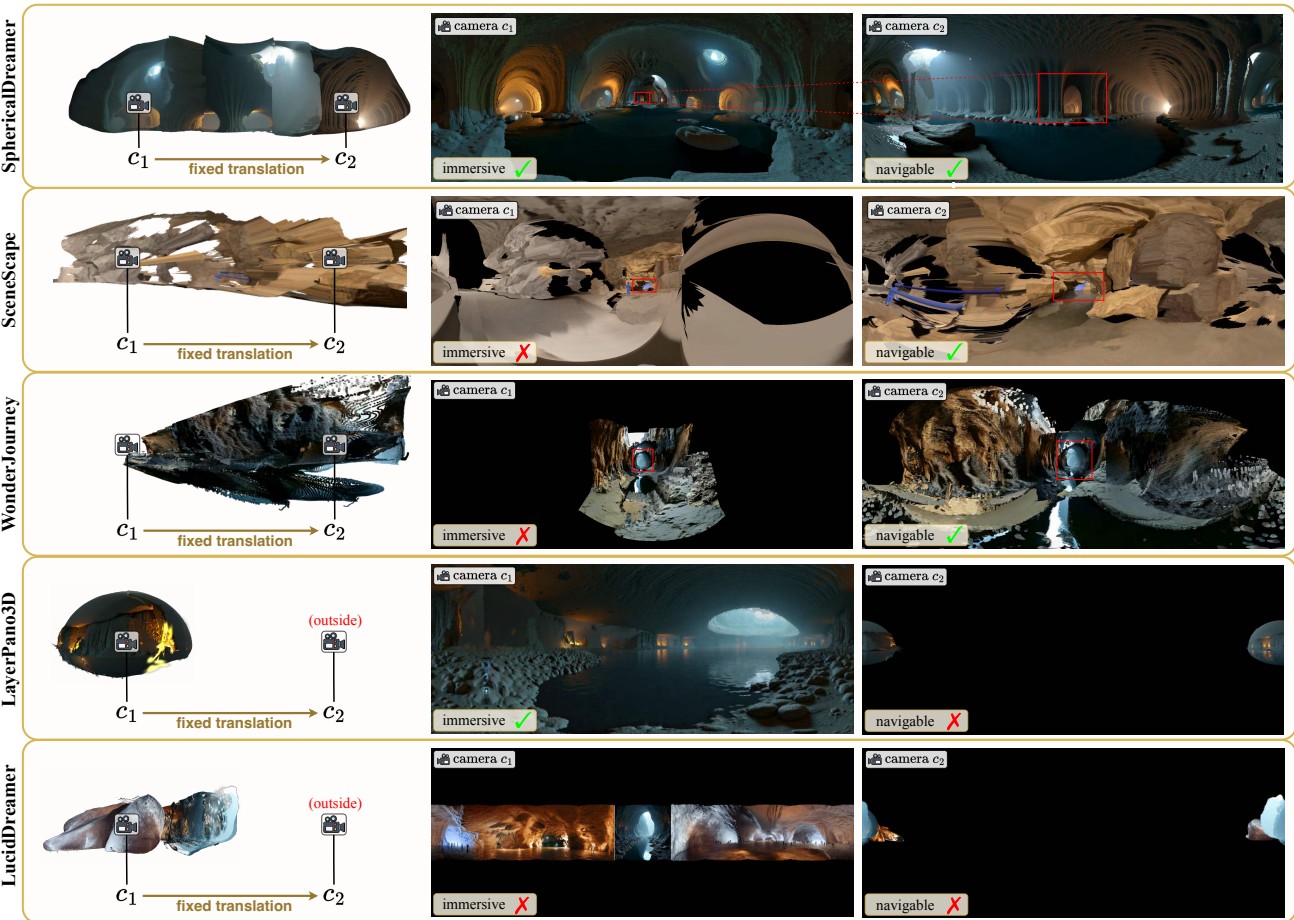

*Figure 5.* **Qualitative comparison over the full** $180° \times 360°$ **field of view across distant camera viewpoints.** Our proposed method SphericalDreamer is the only one to support high quality, full omnidirectional coverage across distant camera viewpoints. In comparison, SceneScape and Wonderjourney renderings are only visually plausible within a restricted field of view, making them non-immersive. For LucidDreamer and LayerPano3D, the second camera lies outside of the 3D world, making them non-navigable.

**Baselines.** We focus on baselines that generate worlds with an explicit 3D representation (e.g. point clouds, meshes, or 3D Gaussian Splatting). Among panorama-based approaches, we compare against LayeredPano3D (Yang et al., 2025). We do not include HoloDreamer (Zhou et al., 2025) and PanoDreamer (Paliwal et al., 2025), as their code is not publicly available. Among iterative completion approaches, we compare against LucidDreamer (Chung et al., 2025), WonderJourney (Yu et al., 2024), and SceneScape (Fridman et al., 2023).

**Experimental details.** In all experiments, we run SphericalDreamer with a world size of $N = 3$ panoramas. The precise off-the-shelf models used in the SphericalDreamer pipeline are listed in Table 3 of the Appendix.

**Evaluation protocol.** Using a set of diverse textual prompts, each baseline is run on each prompt. For baselines requiring both text and an input image, the input image is generated using Flux (Black Forest Labs, 2024) conditioned on the same text prompt. The set of prompts is available in Table 4 of the Appendix. For each generated 3D world, we consider the default camera location (e.g. panorama center for LayeredPano3D). Starting from this reference camera, we construct three types of camera trajectories: (i) rotation only, measuring local immersivity at the starting location; (ii) translation only, measuring navigability within the scene; and (iii) combined rotation and translation, evaluating immersive navigation. Note that prior to evaluation, all 3D worlds are scale-aligned to ensure that translation magnitudes are comparable across methods. For each scene, we sample 20 rotations, 20 translations, and 20 rotation+translation camera poses. We subsequently render the scenes at each camera pose. Note that rendering is done with perspective cameras for the quantitative evaluations.

**Metrics.** As no ground-truth images are available, we rely on non-reference image quality assessment. Specifically, we

*Table 2.* **Quantitative comparison on image quality and coverage.** SphericalDreamer is the only method to achieve full coverage across all camera trajectories, while delivering state-of-the-art image quality. The reported metrics are averaged over all scenes and views.

| Method | Rotation | | Translation | | Rotation & Translation | |
|---|---|---|---|---|---|---|
| | BRISQUE ↓ | Coverage ↑ | BRISQUE ↓ | Coverage ↑ | BRISQUE ↓ | Coverage ↑ |
| SceneScape | 52.50 | 0.796 | 44.32 | 0.960 | 55.91 | 0.724 |
| WonderJourney | 57.36 | 0.556 | 41.31 | 0.998 | 61.68 | 0.404 |
| LayerPano3D | 48.40 | **1.000** | 70.08 | 0.476 | 76.74 | 0.594 |
| LucidDreamer | 62.54 | 0.798 | 65.16 | 0.682 | 64.35 | 0.775 |
| SphericalDreamer | **44.96** | 0.999 | **36.57** | **0.999** | **41.73** | **0.999** |

report the BRISQUE (Mittal et al., 2012) metric to evaluate the quality of the rendered images. To assess immersivity along camera trajectories, we report the scene *coverage*, defined as the proportion of pixels in the rendered image that originate from the 3D world rather than the default black background.

**Results.** Figure 5 qualitatively compares the methods by showing the appearance of the generated worlds from two distant camera locations, using equirectangular renderings to assess immersivity. It showcases that SphericalDreamer is the only method to be simultaneously immersive and navigable.

Table 2 reports quantitative results for image coverage and quality along the three types of camera trajectories. While other methods achieve strong coverage in either rotation or translation, indicating that they are either immersive or navigable, none perform well on both simultaneously. In contrast, SphericalDreamer is the only method that achieves complete coverage when using rotation and translation combined. When only rotation or translation are used, SphericalDreamer also achieves full coverage. This demonstrates that SphericalDreamer is the first method to enable immersive navigation within generated 3D worlds. Furthermore, SphericalDreamer achieves the best BRISQUE scores across all trajectories, demonstrating state-of-the-art image quality.

All in all, SphericalDreamer is the first approach to be simultaneously immersive and navigable, establishing a new state-of-the-art for high quality 3D world generation.

**Additional evaluations.** We complement the evaluation above with additional analyses, fully reported in the Appendix. First, additional qualitative comparisons against baselines confirm that SphericalDreamer is the only method to be simultaneously navigable and immersive (Figures 15 to 19), while four supplementary metrics (CLIP-Score, C-CLIP, CLIP-IQA, and Q-Align) further corroborate its state-of-the-art quality (Table 5). Second, an ablation study shows that removing either Layered Depth Panoramas (LDPs) or Harmonic Blending (HB) causes consistent quality drops (Figure 9 and Table 6). We further conduct dedicated component-level comparisons: our LDP construction yields

cleaner background panoramas than those of LayerPano3D and 3D Photography (Section C.5 and Figures 12 to 14); HB produces smoother depth transitions than simpler alternatives (Section C.6 and Figures 4 and 7); and our depth maps are more accurate and exhibit fewer artifacts than those of four dedicated $360°$ depth estimation models (Section C.7, Figure 11, and Table 9). Finally, quality metrics remain stable as world size grows from $N = 3$ to 7 (Table 7), and a detailed runtime breakdown shows that a full world with $N = 3$ panoramas is generated in approximately 40 minutes on a single A100 GPU (Table 10).

### 4.1. Limitations

Despite its strong performance, SphericalDreamer is less effective with scenes requiring precise planar geometry, such as urban or indoor environments. This limitation arises from the use of spherical imagery, where depth estimation is less accurate than for perspective images, leading to curvature artifacts in the reconstructed geometry. Advances in depth estimation for panoramas would help alleviate these issues for SphericalDreamer and other panorama-based methods.

## 5. Conclusion

We present SphericalDreamer, a framework for generating large-scale, immersive, and navigable 3D worlds from textual prompts. By extending panorama-based generation to a multi-panorama setting and introducing a generative fusion mechanism to connect them, our method overcomes a key limitation of prior approaches: the inability to simultaneously achieve full omnidirectional view coverage and long-range navigability.

Through extensive qualitative and quantitative evaluations, we show that SphericalDreamer consistently outperforms existing panorama-based and iterative completion methods, producing 3D environments that are both visually immersive and explorable over large spatial extents. Our results demonstrate that panoramic representations, when combined with generative fusion, offer a scalable and effective foundation for 3D world generation.

## Impact Statement

This work introduces a method for generating immersive and navigable 3D environments from text, with potential applications in areas such as virtual reality, simulation, and digital content creation. By reducing the effort required to produce large-scale 3D scenes, it may support research and creative workflows that rely on virtual 3D environments. As the method builds upon pretrained 2D generative models, any limitations or biases present in these components may be reflected in the generated 3D content. We view this work as a research contribution and encourage responsible use with appropriate oversight.

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

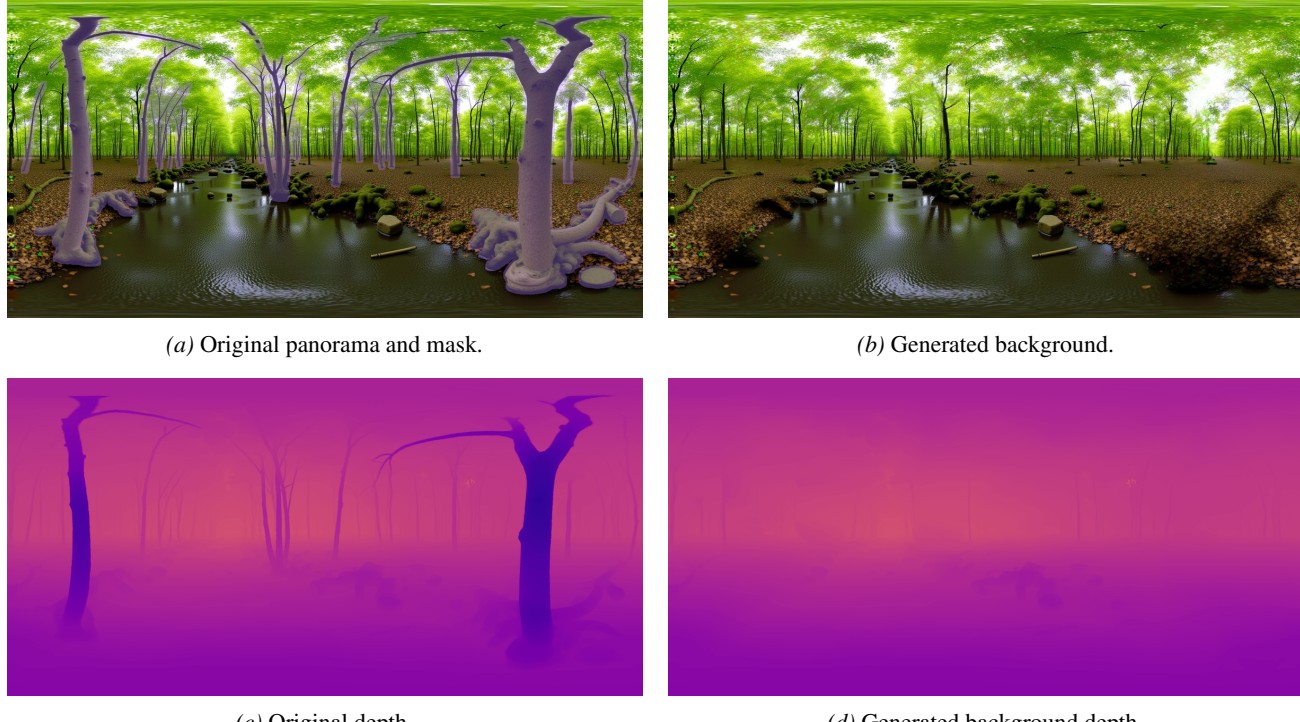

*(a)* Original panorama and mask.  *(b)* Generated background.

*(c)* Original depth.  *(d)* Generated background depth.

*Figure 6.* **Layered depth panorama (LDP).** Foreground regions (purple mask) of the panorama (a) are removed and inpainted to produce a background image (b). The original depth (c) is used to complete the background depth (d) by taking its maximum along each row.

## A. Layered Depth Image Construction

Given an RGB panorama $I \in \mathbb{R}^{H \times W \times 3}$ and its corresponding depth map $D \in \mathbb{R}^{H \times W}$, our goal is to construct a background RGB–D layer $(I^{\mathrm{bg}}, D^{\mathrm{bg}})$ in which regions corresponding to foreground elements in the original panorama $(I, D)$ are replaced with plausible background content and depth. Figure 6 illustrates the resulting LDP on a representative scene.

**Foreground mask computation.** To identify foreground objects, we leverage the fact that foreground regions typically induce strong depth discontinuities with respect to their surroundings. We first compute depth edges using the Canny edge detector (Canny, 1986), yielding a binary edge mask $C$. In parallel, we infer a set of semantic segmentation masks $\{S_k\}$ using Segment Anything (SAM, Kirillov et al. (2023)). For each mask $S_k$, we extract its boundary $B_k$ and estimate an outward-pointing normal $\mathbf{n}(x)$ at each boundary point $x \in B_k$ from the spatial gradient of $S_k$. We then evaluate how strongly the segmented object lies in front of its surrounding regions by measuring depth variations across the boundary, restricted to boundary points aligned with depth edges. Specifically, for each $x \in B_k \wedge C$, we sample the depth map at two locations offset by a small distance $\varepsilon > 0$ along the normal direction: one inside the mask and one outside. This yields a signed depth difference

$$\Delta d(x) = d(x + \varepsilon \mathbf{n}(x)) - d(x - \varepsilon \mathbf{n}(x)),$$

where $d$ is a continuous depth functional induced by the discrete depth map $D$, allowing depth values to be queried at arbitrary spatial locations $x$ via bilinear interpolation.

When the object represented by $S_k$ is in front of its surroundings, the depth outside the boundary is larger than the depth inside, resulting in $\Delta d(x) > 0$. We aggregate these measurements along the boundary to obtain a foreground score

$$s_k = \frac{1}{|B_k \wedge C|} \int_{B_k \wedge C} \Delta d(x) \, \mathrm{d}x,$$

which is approximated in practice by averaging over valid boundary samples. Masks exhibiting consistently positive depth differences with large magnitude are thus identified as foreground candidates. Finally, we select foreground candidates by

thresholding the score $s_k$ with an empirically chosen threshold $t$. All masks satisfying $s_k > t$ are retained, and the final foreground mask is obtained by accumulating the selected candidates via a pixel-wise logical OR:

$$M^{\text{fg}} = \bigvee_{k \in \mathcal{K}} S_k,$$

where $\mathcal{K}$ denotes the set of indices corresponding to the selected foreground masks.

**Foreground-Aware Background Inpainting.** Following the procedure implemented by Yang et al. (2025), we inpaint background content upon an input panorama $I$ and its estimated foreground mask $M^{\text{fg}}$. Specifically, we apply a two-stage inpainting strategy to generate a background panorama $I^{\text{bg}}$. First, a coarse inpainting model (LaMa, Suvorov et al. (2022)) is used to complete the masked regions at reduced resolution, producing a coarse background estimate $I^{\text{bg}-\text{coarse}}$. We then automatically caption $I^{\text{bg}-\text{coarse}}$ using an image captioning model (LLaVa, Liu et al. (2023)) to obtain a textual description $\hat{p}$. Finally, a high-quality diffusion-based inpainting model (Flux, Black Forest Labs (2024)) refines the coarse completion conditioned on $\hat{p}$, yielding the final background panorama:

$$I^{\text{bg}-\text{coarse}} = \text{INPAINT-LAMA}(I, M^{\text{fg}}), \qquad \hat{p} = \text{CAPTION}(I^{\text{bg}-\text{coarse}}), \qquad I^{\text{bg}} = \text{INPAINT-FLUX}(I^{\text{bg}-\text{coarse}}, M^{\text{fg}}; \hat{p}). \tag{11}$$

This procedure produces a background panorama $I^{\text{bg}}$ with plausible content for the masked regions.

Intuitively, the two-stage inpainting strategy decouples background texture synthesis from high-quality refinement. The first model produces a coarse completion containing exclusively background content with coherent textures. The second model, which would otherwise introduce salient foreground objects, is instead conditioned on this coarse background and applied with reduced inpainting strength. This improves visual quality while ensuring that only background elements are generated.

**Panoramic hull for depth estimation** Let $D \in \mathbb{R}^{H \times W}$ denote the depth of a panorama. For each elevation index $y$, we compute the farthest valid depth across azimuth,

$$h(y) = \max_x d(y, x),$$

and broadcast this profile over all columns to obtain a row-constant depth map $D^{\text{bg}}(y, x) = h(y)$. This defines a *panoramic hull*: a smooth enclosing surface that captures the global background extent of the scene at each elevation while suppressing foreground geometry. We use $D^{\text{bg}}$ to backproject the newly estimated background panorama $I^{\text{bg}}$ into 3D, ensuring that the resulting background geometry encloses the existing content.

## B. Harmonic Blending for Depth Alignment

Inspired by classical Laplacian mesh editing and harmonic surface deformation (Sorkine et al., 2004; Yu et al., 2004; Botsch et al., 2010; Jacobson et al., 2010), we introduce harmonic blending, a pipeline that fuses newly generated geometry into the existing 3D world. The core idea is to treat the estimated depth as a deformable point cloud, and to smoothly deform it so that it exactly matches the reliable rendered depth along their shared boundary. This allows us to reconcile a locally plausible but globally misaligned depth map with a partially defined, geometrically grounded depth map, yielding an artifact-free fusion in 3D.

**Input and notations.** We are given two depth maps of identical spatial resolution, $D^{\text{r}}, D^{\text{est}} \in \mathbb{R}^{H \times W}$, defined over the same image grid. Here, $D^{\text{r}}$ denotes the rendered, geometrically reliable depth obtained from existing 3D geometry, while $D^{\text{est}}$ is a monocular depth estimate predicted from newly synthesized image content. We are also given a binary validity mask $M^{\text{r}} \in \{0, 1\}^{H \times W}$, indicating the spatial support on which $D^{\text{r}}$ is defined and trusted.

**Domain partition.** The mask $M^{\text{r}}$ directly induces a partition of the depth domain. Pixels where $M^{\text{r}} = 1$ correspond to grounded geometry, where the rendered depth $D^{\text{r}}$ must be preserved. Its complement $1 - M^{\text{r}}$ identifies synthesized regions, where only the estimated depth $D^{\text{est}}$ is available and must be smoothly deformed to blend with existing geometry.

To couple both regions, we define a narrow boundary mask $\partial M^{\text{r}} \subset M^{\text{r}}$, obtained by a one-pixel erosion of $M^{\text{r}}$, or equivalently as the contour of the complementary mask $1 - M^{\text{r}}$. By construction, both depth maps $D^{\text{r}}$ and $D^{\text{est}}$ are defined on $\partial M^{\text{r}}$.

This boundary mask is used to enforce equality between the two depth maps, thereby anchoring the deformation of $D^{\text{est}}$ to the grounded depth $D^{\text{r}}$ along a narrow transition band and enabling the geometric blend.

**3D back-projection.**  Using the spherical back-projection operator $\Pi_{\mathbb{S}}^{-1}$ and the camera pose $\mathbf{T}$, we lift the depth maps into 3D while respecting the mask-based partition.

Deformable points are obtained by back-projecting the estimated depth on the complementary region:

$$P \;=\; \Pi_{\mathbb{S}}^{-1}\big(D^{\text{est}},\, \mathbf{T},\, 1 - M^{\text{r}}\big).$$

On the boundary mask $\partial M^{\text{r}}$, both depth maps are defined. We therefore distinguish two types of boundary points:

$$P_{\partial} \;=\; \Pi_{\mathbb{S}}^{-1}\big(D^{\text{est}},\, \mathbf{T},\, \partial M^{\text{r}}\big), \qquad P_{\partial}^{\text{target}} \;=\; \Pi_{\mathbb{S}}^{-1}\big(D^{\text{r}},\, \mathbf{T},\, \partial M^{\text{r}}\big).$$

By construction, points in $P_{\partial}$ and $P_{\partial}^{\text{target}}$ are in one-to-one correspondence through the shared mask $\partial M^{\text{r}}$. As $P_{\partial}^{\text{target}}$ represents the trusted geometry, enforcing $P_{\partial}$ to match $P_{\partial}^{\text{target}}$ provides boundary conditions that guide the deformation of the synthesized geometry $P$ and ensure geometric continuity with the grounded depth.

**Graph-based harmonic deformation.**  Considering the triplet $(P, P_{\partial}, P_{\partial}^{\text{target}})$, we construct the set of deformable point $\mathcal{P}$ for which we seek a deformation field $\mathcal{U}$ :

$$\mathcal{P} = \begin{bmatrix} P \\ P_{\partial} \end{bmatrix} \in \mathbb{R}^{N \times 3}, \qquad \mathcal{U} = \begin{bmatrix} U \\ U_{\partial} \end{bmatrix} \in \mathbb{R}^{N \times 3},$$

where the application of $\mathcal{U}$ yields the deformed points $\mathcal{P}'$ :

$$\mathcal{P}' = \mathcal{P} + \mathcal{U}, \qquad \mathcal{P}' = \begin{bmatrix} P' \\ P'_{\partial} \end{bmatrix} \in \mathbb{R}^{N \times 3}.$$

The deformation is constrained along the interface by enforcing exact alignment with the grounded geometry:

$$U_{\partial} = P_{\partial}^{\text{target}} - P_{\partial}.$$

Outside the boundary, the deformation is determined by a local rigidity prior. To this end, we construct a weighted $k$-nearest-neighbor graph over $\mathcal{P}$, with edge weights $w_{ij}$ inversely proportional to the Euclidean distance between the points $\mathcal{P}_i$ and $\mathcal{P}_j$. Let $L$ denote the associated (unnormalized) graph Laplacian. We compute the displacement field as the solution of the following constrained energy minimization:

$$\mathcal{U} = \arg\min_{\mathcal{V}} E(\mathcal{V}), \qquad E(\mathcal{V}) = \sum_{(i,j)} w_{ij} \, \|\mathcal{V}_i - \mathcal{V}_j\|_2^2 = \text{tr}(\mathcal{V}^{\top} L \mathcal{V}),$$

subject to the boundary alignment constraints defined above. This yields a sparse linear system, which we solve using standard sparse solvers.

Intuitively, the Dirichlet energy penalizes differences in displacement between neighboring points, encouraging locally coherent motion across the graph. As a result, nearby points tend to move together with similar displacements, which approximately preserves local relative configurations and prevents shearing or depth seams. This behavior mirrors classical harmonic and Laplacian deformation in mesh editing, where boundary constraints are smoothly propagated into unconstrained regions (Sorkine et al., 2004; Yu et al., 2004; Botsch et al., 2010; Jacobson et al., 2010).

The resulting deformed point set $\mathcal{P}'$ exactly satisfies the boundary condition while locally preserving the geometry of $\mathcal{P}$.

**Output.**  After this step, we obtain the point cloud $P'$, defined as $\mathcal{P}'$ with boundary points removed. By construction, it aligns perfectly with the already existing geometry defined by $D^{\text{r}}$.

More formally, we can derive a depth map $D^{\text{blend}}$ by reprojecting the deformed point set $P'$ into the spherical image domain associated with the camera pose $\mathbf{T}$ :

$$D^{\text{blend}} \;=\; \Pi_{\mathbb{S}}\big(P',\, \mathbf{T}\big).$$

*Table 3.* Overview of the models and tools used in the SphericalDreamer pipeline.

| Task | Models | Reference |
|---|---|---|
| Panorama generation | FLUX.1 [dev]
with LoRA for panorama | Black Forest Labs (2024)
Yang et al. (2025) |
| Panorama inpainting | Fine inpainting: FLUX.1 Fill [dev]
Coarse inpainting: LaMA
Auto-captioning: LLaVA | Black Forest Labs (2024)
Suvorov et al. (2022)
Liu et al. (2023) |
| Depth estimation | 360MonoDepth
with DepthAnythingV2 as backbone | Rey–Area et al. (2022)
Yang et al. (2024) |
| Semantic segmentation | Segment-Anything | Kirillov et al. (2023) |
| Point-cloud rendering | Blender | Blender Foundation (2018) |

By construction, $D^{\mathrm{blend}}$ is defined exactly on the synthesized regions of the panorama, i.e., on the validity region of the mask $1 - M^{\mathrm{r}}$, where the original rendered depth $D^{\mathrm{r}}$ was undefined. On the regions where the rendered depth is valid, we explicitly preserve the grounded geometry by assigning $D^{\mathrm{blend}} \equiv D^{\mathrm{r}}$. In fine, $D^{\mathrm{blend}}$ is defined everywhere on the image domain and is free of geometric or alignment discontinuities.

**Visual comparisons.** Figure 4 provides a visual comparison between naive depth composition and harmonic blending. The naive depth map is obtained by directly combining the two inputs: we set $D^{\mathrm{naive}} \equiv D^{\mathrm{r}}$ on the validity support of $M^{\mathrm{r}}$, and $D^{\mathrm{naive}} \equiv D^{\mathrm{est}}$ on the complementary region corresponding to synthesized content. While simple, this strategy introduces visible depth discontinuities at the interface between the two regions. In contrast, harmonic blending produces a seamless depth map, eliminating geometric discontinuities across the transition.

Figure 7 illustrates harmonic blending in 3D on a toy example, highlighting the resulting point cloud deformations. In this illustrative setting, additional fixed points are introduced by treating them as extra boundary constraints.

# C. Additional Results

This appendix presents in detail the additional experiments that are only briefly summarised in the corresponding chapter. We first report implementation details of our pipeline (Section C.1), before presenting additional quantitative and qualitative comparisons (Section C.2) and an ablation study of LDP and Harmonic Blending (Section C.3). We then study how our pipeline behaves when the world size grows (Section C.4), compare our Layered Depth Panorama construction with prior layered representations (Section C.5), and analyse Harmonic Blending in isolation on the task of depth completion (Section C.6). Finally, we report a dedicated geometry evaluation against $360°$ depth estimation baselines on Replica2K (Section C.7) and provide a runtime breakdown of the pipeline (Section C.8).

## C.1. Implementation Details

Table 3 lists the off-the-shelf models used at each stage of the SphericalDreamer pipeline, and Table 4 reports the full set of textual prompts used to generate the 3D worlds in our quantitative and qualitative evaluations.

## C.2. Additional Quantitative and Qualitative Comparisons

We complement the main quantitative evaluation with four additional metrics computed on all baselines under combined rotation and translation trajectories. CLIP-Score and C-CLIP measure prompt alignment and cross-view semantic consistency, respectively, while CLIP-IQA and Q-Align measure rendering quality; since geometric artifacts typically degrade visual quality, these metrics also reflect underlying geometric consistency. SphericalDreamer achieves the best score on every metric (Table 5), further corroborating its state-of-the-art quality.

Additional qualitative comparisons against all baselines are presented in Figures 15 to 19. Across all scenes and prompts, SphericalDreamer is consistently the only method to be simultaneously navigable and immersive.

*Table 4.* Text prompts used in our experiments for generating 3D environments.

| Scene Name | Text Prompt |
|---|---|
| *cave river* | A large-scale subterranean cave inspired by the Phantom of the Opera underground river, featuring an irregular cavern corridor extending forward and behind the observer, rounded organic rock formations, a dark slow-moving river with reflective surface and subtle ripples, wet rocky banks, smoothed stalactites and stalagmites, faint mist above the water, dim atmospheric lighting with warm reflections, deep shadows, and a cinematic gothic mood. |
| *jungle* | A dense rainforest understory extending forward and behind the observer, with thick overlapping vegetation, large deep-green leaves, moist ground covered in roots and organic debris, an overcast sky filtering light through the canopy, soft diffused illumination, and a humid, lush tropical atmosphere. |
| *underwater world* | A wide coral reef canyon extending forward and behind the observer, with smooth rock walls covered in coral and marine growth, vibrant yet softened turquoise, coral pink, and sandy beige tones, filtered underwater lighting with soft light rays, floating particles, and a calm immersive oceanic environment. |
| *martian desert* | A Martian landscape of rust-red and ochre soil scattered with dark basalt rocks and patches of muted green and purple alien vegetation, under a dusty salmon-colored sky. |
| *grass field* | A wide open rolling field of lush green grass inspired by *The Sound of Music*, with gentle hills extending forward and behind the observer, smooth terrain, thick healthy grass, subtle wildflowers, an overcast sky, soft diffused daylight, a distant tree line, and a peaceful cinematic pastoral atmosphere. |
| *desolated landscape* | A desolate Upside Down-inspired landscape extending forward and behind the observer, with dark damp ground covered in tangled root-like growth, floating ash-like particles, twisted vegetation, murky fog, muted blue-gray lighting, wet reflective patches, slimy textures, and an immersive cinematic horror atmosphere. |

*Table 5.* **Additional quantitative comparison on 3D world generation.** Text alignment and global semantic consistency are measured with CLIP-Score and C-CLIP, while rendering quality is measured with CLIP-IQA and Q-Align. Scores are reported under combined rotation and translation trajectories. SphericalDreamer achieves the best score on every metric.

| Method | CLIP-Score ↑ | C-CLIP ↑ | CLIP-IQA ↑ | Q-Align ↑ |
|---|---|---|---|---|
| LucidDreamer | 0.2988 | 0.6599 | 0.6237 | 1.8908 |
| LayerPano3D | 0.2639 | 0.5787 | 0.5077 | 1.7943 |
| WonderJourney | 0.2570 | 0.6149 | 0.5006 | 1.6017 |
| SceneScape | 0.2802 | 0.7866 | 0.3805 | 1.7231 |
| **SphericalDreamer** | **0.3325** | **0.8433** | **0.7014** | **2.3088** |

## C.3. Ablation Study

We ablate the two key components of our pipeline by (i) removing LDP entirely and (ii) replacing Harmonic Blending (HB) with naïve blending, bilinear interpolation and diffusion-based depth inpainting (Liu et al., 2024, InFusion). Qualitatively (Figure 9), replacing HB causes geometry artifacts in transition regions, while removing LDP introduces disocclusion artifacts with unfilled regions; the full model produces complete and coherent environments. Quantitatively (Table 6), both ablations cause consistent drops across all metrics: HB improves completeness and visual quality, and LDP is critical for handling occlusions and maintaining consistency.

## C.4. Varying the World Size

We provide additional experiments in which we generate worlds of varying size, sweeping over $N \in \{3, 4, 5, 6, 7\}$ fused panoramas.

Quantitatively (Table 7), we evaluate six complementary metrics: BRISQUE, Coverage, CLIP-Score, C-CLIP, CLIP-IQA and Q-Align. All six scores remain stable as $N$ increases, demonstrating that our generated worlds maintain similar levels of quality, global semantic consistency and prompt alignment as the world grows.

*Table 6.* **Ablation of Layered Depth Panoramas (LDP) and Harmonic Blending (HB).** We remove LDP from the pipeline and replace HB with three simpler alternatives: naïve blending, bilinear interpolation and diffusion-based depth inpainting (Liu et al., 2024, InFusion) Both components are required to reach the best scores on every metric.

| Setting | BRISQUE ↓ | Coverage ↑ | CLIP-Score ↑ | C-CLIP ↑ | CLIP-IQA ↑ | Q-Align ↑ |
|---|---|---|---|---|---|---|
| no LDP | 45.8937 | 0.9652 | 0.3371 | 0.8459 | 0.7358 | 2.1935 |
| no HB (naïve instead) | 47.1084 | 0.9242 | 0.3253 | 0.8001 | 0.6503 | 1.9953 |
| no HB (inpainting instead) | 44.3742 | 0.9314 | 0.3272 | 0.8099 | 0.5976 | 2.0780 |
| no HB (interpolation instead) | 45.4247 | 0.9835 | 0.3377 | 0.8496 | 0.7505 | 2.2595 |
| **Ours (LDP + HB)** | **43.4709** | **0.9993** | **0.3418** | **0.8608** | **0.7582** | **2.3660** |

*Table 7.* **Effect of the world size $N$ on generation quality.** Quantitative evaluation as we increase the number of fused panoramas from $N = 3$ to $N = 7$. All metrics remain stable, indicating that SphericalDreamer preserves quality, coverage and global semantic consistency as the world grows.

| Setting | BRISQUE ↓ | Coverage ↑ | CLIP-Score ↑ | C-CLIP ↑ | CLIP-IQA ↑ | Q-Align ↑ |
|---|---|---|---|---|---|---|
| $N=3$ | 43.4683 | 0.9993 | 0.3418 | **0.8608** | 0.7582 | **2.3662** |
| $N=4$ | 42.2766 | 0.9993 | 0.3418 | 0.8471 | 0.7387 | 2.3057 |
| $N=5$ | 41.5480 | 0.9994 | 0.3403 | 0.8447 | 0.7747 | 2.3088 |
| $N=6$ | 42.1268 | **0.9995** | 0.3400 | 0.8499 | 0.7777 | 2.3120 |
| $N=7$ | **41.5514** | 0.9994 | **0.3424** | 0.8453 | **0.7913** | 2.3490 |

## C.5. LDP Qualitative Comparisons

We compare the LDP construction of SphericalDreamer with two prior layered representations, namely those used in LayerPano3D (Yang et al., 2025) and 3D Photography (Shih et al., 2020, 3DP). For each method, we show the input panorama overlaid with the detected foreground (left) and the inpainted background obtained after removing the detected foreground (right).

As shown in the figures, SphericalDreamer produces more realistic background panoramas without artifacts, owing to more accurate foreground segmentation and occlusion-aware mask estimation.

## C.6. Harmonic Blending: Comparisons with Other Alternatives

Our ablation study (Section C.3) already establishes that Harmonic Blending is critical inside the SphericalDreamer pipeline. We additionally study Harmonic Blending in isolation on the specific task of *depth completion*, which offers a more direct understanding of why it is more effective than simpler alternatives.

In this protocol we mask a portion of a reference depth map and reconstruct the missing region using each candidate method: bilinear interpolation, diffusion-based depth inpainting (Liu et al., 2024, InFusion) and our Harmonic Blending (HB). We evaluate the reconstructions with two metrics:

- The **Depth Transition Score** is the average absolute difference in depth values across pixels on either side of the boundary between known and reconstructed regions. Lower values indicate smoother transitions.

- The **Transition Region MAE** (Depth Estimation Error) measures the difference between predicted and ground-truth depth values within a narrow band along the blending boundary, inside the missing regions. This band captures transition areas where geometric inconsistencies are most likely to occur.

Qualitatively (Figure 10), Harmonic Blending provides the smoothest transition between known and reconstructed regions, without causing any artifacts. Quantitatively (Table 8), our approach achieves the best Depth Transition Score, indicating the smoothest transition between known and unknown depth, and the lowest Transition Region MAE, indicating closer agreement with ground-truth depth and more coherent geometry in these regions. These results confirm that Harmonic Blending ensures seamless alignment in overlapping areas.

*Table 8.* **Harmonic Blending vs. simpler alternatives on depth completion.** We mask a portion of a reference depth map and reconstruct it with each method. The *Depth Transition Score* is the average absolute depth difference across the boundary between known and reconstructed regions (lower is smoother). The *Transition Region MAE* measures the MAE with respect to ground-truth depth inside a narrow band along the boundary. Our method achieves the best scores on both metrics by a wide margin.

| Method | Depth Transition Score ↓ | Transition Region MAE ↓ |
|---|---|---|
| Interpolation | 0.00389 | 0.03067 |
| Depth inpainting (Liu et al., 2024, InFusion) | 0.09529 | 0.09177 |
| **Harmonic Blending (Ours)** | **0.00328** | **0.00079** |

*Table 9.* **Quantitative depth evaluation on Replica2K.** We compare the depth maps produced inside our pipeline (SphericalDreamer via 360MonoDepth) with four dedicated 360° depth estimators: BiFuseV2 (Wang et al., 2023a), EGFormer (Yun et al., 2023), HoHoNet (Sun et al., 2021) and UniFuse (Jiang et al., 2021). Our approach consistently outperforms all baselines on every metric.

| Model | AbsRel ↓ | RMSE ↓ | SI-RMSE ↓ | $\delta < 1.25$ ↑ | $\delta < 1.25^2$ ↑ | $\delta < 1.25^3$ ↑ |
|---|---|---|---|---|---|---|
| BiFuseV2 | 1.0077 | 1.8958 | 1.0858 | 0.1736 | 0.3368 | 0.4906 |
| EGFormer | 0.8048 | 1.6097 | 0.8338 | 0.2107 | 0.3952 | 0.5744 |
| HoHoNet | 1.1524 | 2.0839 | 1.1094 | 0.1711 | 0.3301 | 0.4723 |
| UniFuse | 1.0445 | 1.9659 | 1.1372 | 0.1738 | 0.3250 | 0.4706 |
| **SphericalDreamer** | **0.1605** | **0.6008** | **0.2039** | **0.7363** | **0.9317** | **0.9819** |

## C.7. Geometry Evaluation

We evaluate the geometry of our generated 3D worlds via two complementary approaches: direct depth assessment, and indirect evaluation through perceptual image quality metrics.

**Evaluation via predicted depth.** We compare the depth maps produced inside SphericalDreamer (obtained via 360MonoDepth (Rey–Area et al., 2022)) with four dedicated 360° depth estimation models: BiFuseV2 (Wang et al., 2023a), EGFormer (Yun et al., 2023), HoHoNet (Sun et al., 2021) and UniFuse (Jiang et al., 2021).

Qualitatively (Figure 11), our depth maps are the most accurate and present the fewest artifacts compared to other approaches.

Quantitatively (Table 9), we report comparisons on the Replica2K dataset using standard depth evaluation metrics: Absolute Relative Error (AbsRel), Root Mean Squared Error (RMSE) and Scale-Invariant RMSE (SI-RMSE). We also include the $\delta$-accuracy thresholds, which measure the proportion of pixels whose predicted depth falls within increasing error margins ($1.25$, $1.25^2$, $1.25^3$, corresponding to approximately $25\%$, $56\%$ and $95\%$ relative error). Our approach consistently outperforms all baselines across every evaluated metric.

**Evaluation via image quality metrics.** In addition to depth-based evaluation, the CLIP-IQA and Q-Align scores reported in Table 5 and Section C.2 provide an indirect assessment of geometry quality, as geometric artifacts typically degrade visual quality. Our method achieves the best results on both metrics under combined rotation and translation, further supporting the quality of the underlying geometry.

## C.8. Runtime Report

We report in Table 10 the total runtime of our pipeline and the per-component breakdown for $N \in \{2, 3, 4\}$ on a single NVIDIA A100 GPU.

Generating a 3D world with $N = 3$ panoramas and rendering a video trajectory takes approximately $40$ minutes. Generation time scales approximately linearly with $N$.

*Table 10.* **Runtime breakdown of the SphericalDreamer pipeline** for different world sizes on a single NVIDIA A100 GPU. We report the wall-clock time and the percentage of the total runtime spent in each stage for $N = 2, 3, 4$. Total runtime scales approximately linearly with $N$; Harmonic Blending and LDP Inpainting dominate the total budget.

| Stage | $N=2$ Time | % | $N=3$ Time | % | $N=4$ Time | % |
|---|---|---|---|---|---|---|
| **Total pipeline** | **23m 42s** | **100.0** | **39m 19s** | **100.0** | **55m 08s** | **100.0** |
| **Panorama Generation** | 3m 55s | 16.5 | 5m 27s | 13.9 | 6m 37s | 12.0 |
|    Image Generation | 2m 49s | 11.9 | 3m 53s | 9.9 | 4m 33s | 8.3 |
|    Depth Estimation | 1m 04s | 4.5 | 1m 30s | 3.8 | 1m 59s | 3.6 |
| **LDP Inpainting** | 4m 17s | 18.1 | 5m 46s | 14.7 | 7m 49s | 14.2 |
|    SAM Foreground Masking | 40s | 2.8 | 57s | 2.4 | 1m 21s | 2.4 |
|    LAMA Inpainting | 36s | 2.5 | 26s | 1.1 | 38s | 1.1 |
|    FLUX Inpainting | 2m 40s | 11.3 | 3m 56s | 10.0 | 5m 14s | 9.5 |
|    Depth Inpainting | 0s | 0.0 | 0s | 0.0 | 0s | 0.0 |
| **Generative Fusion** | 3m 01s | 12.7 | 6m 07s | 15.6 | 9m 00s | 16.3 |
|    Point Cloud Rendering | 58s | 4.1 | 1m 56s | 4.9 | 3m 01s | 5.5 |
|    FLUX Inpainting | 1m 27s | 6.1 | 2m 58s | 7.5 | 4m 17s | 7.8 |
|    Depth Estimation | 36s | 2.5 | 1m 13s | 3.1 | 1m 42s | 3.1 |
| **LDP Inpainting (interm.)** | 2m 50s | 12.0 | 4m 42s | 12.0 | 6m 09s | 11.2 |
|    SAM Foreground Masking | 25s | 1.8 | 39s | 1.7 | 55s | 1.7 |
|    LAMA Inpainting | 34s | 2.4 | 34s | 1.4 | 36s | 1.1 |
|    FLUX Inpainting | 1m 39s | 7.0 | 3m 05s | 7.8 | 4m 10s | 7.6 |
|    Depth Inpainting | 0s | 0.0 | 0s | 0.0 | 0s | 0.0 |
| **Harmonic Blending** | 7m 23s | 31.2 | 14m 41s | 37.3 | 21m 43s | 39.4 |
| **Video Rendering** | 2m 16s | 9.6 | 2m 36s | 6.6 | 3m 50s | 7.0 |

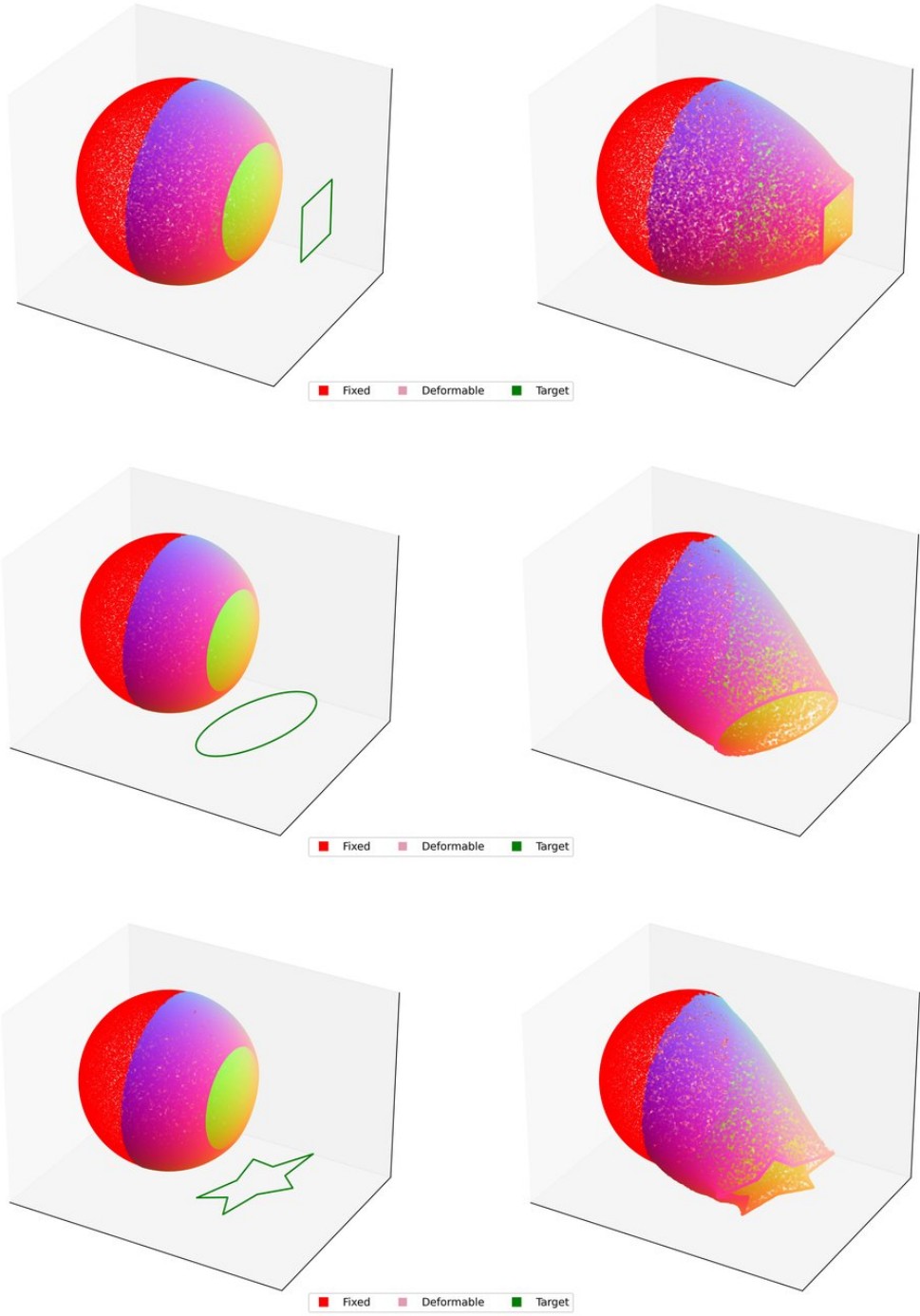

*Figure 7.* **Harmonic Blending on toy examples.** Visualization of deformation results using harmonic blending. The left column shows the deformation setup, with fixed points highlighted in red and the target locations in green. The right column displays the resulting deformed point cloud.

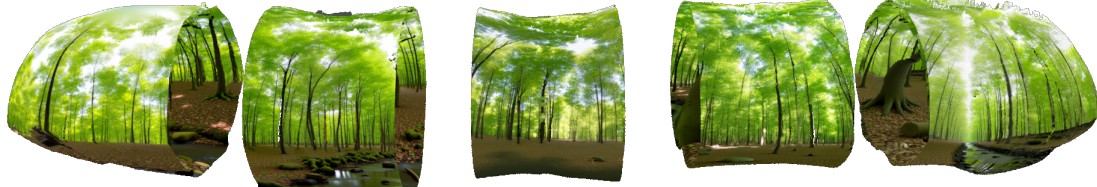

*Figure 8.* **Partial 3D world.** After the first stage of SphericalDreamer, the spherical building blocks can already be assembled to form a 3D world $\mathcal{W}^{\text{partial}}$. However, this intermediate construction still contains missing regions that must be completed in subsequent stages.

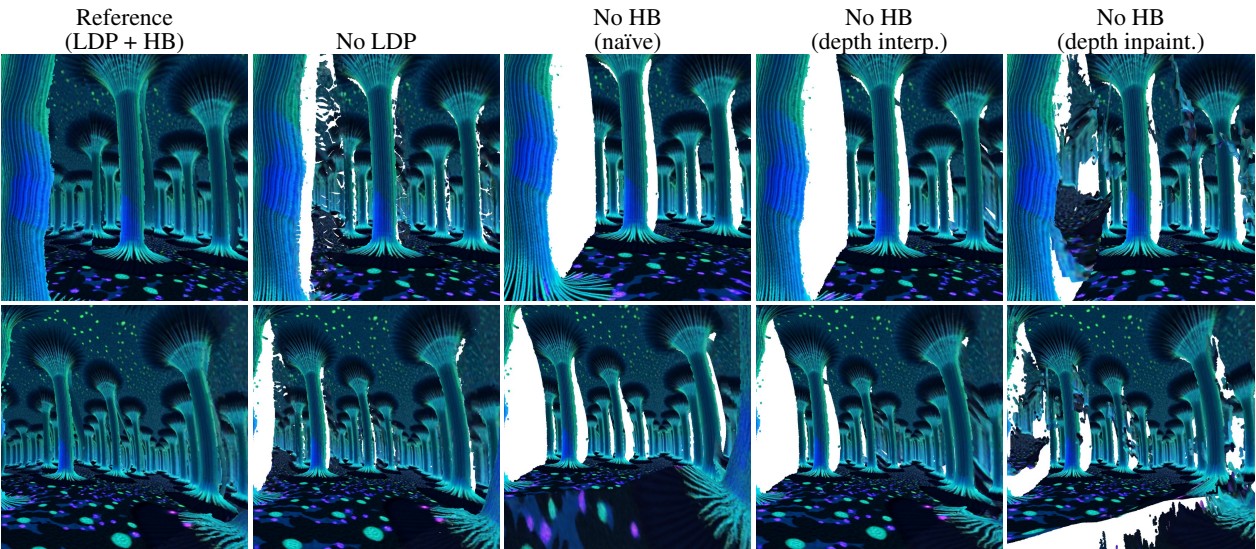

*Figure 9.* **Ablation of LDP and Harmonic Blending.** Comparisons of frames rendered with our full pipeline (left) compared against variants that remove LDP, or that replace Harmonic Blending with naïve blending, bilinear depth interpolation, and diffusion-based depth inpainting (Liu et al., 2024, InFusion). Replacing HB causes geometry artifacts in transition regions; removing LDP introduces disocclusion artifacts with unfilled regions. The full model produces complete and coherent environments.

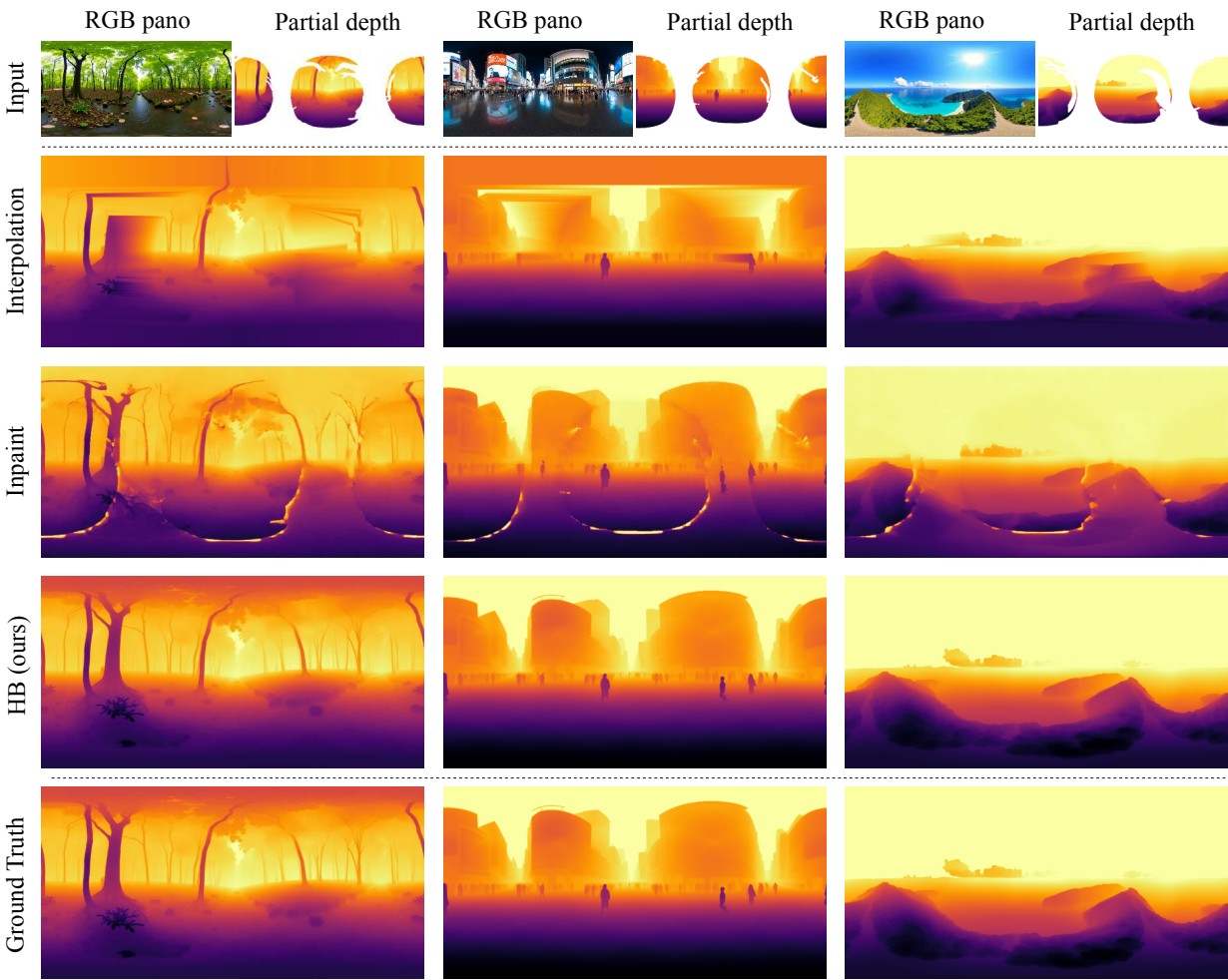

*Figure 10.* **Qualitative comparison of depth completion methods.** Masked regions of a reference depth map reconstructed via bilinear interpolation, diffusion-based depth inpainting (Liu et al., 2024, InFusion) and our Harmonic Blending. Harmonic Blending provides the smoothest transition between known and reconstructed regions, without artifacts.

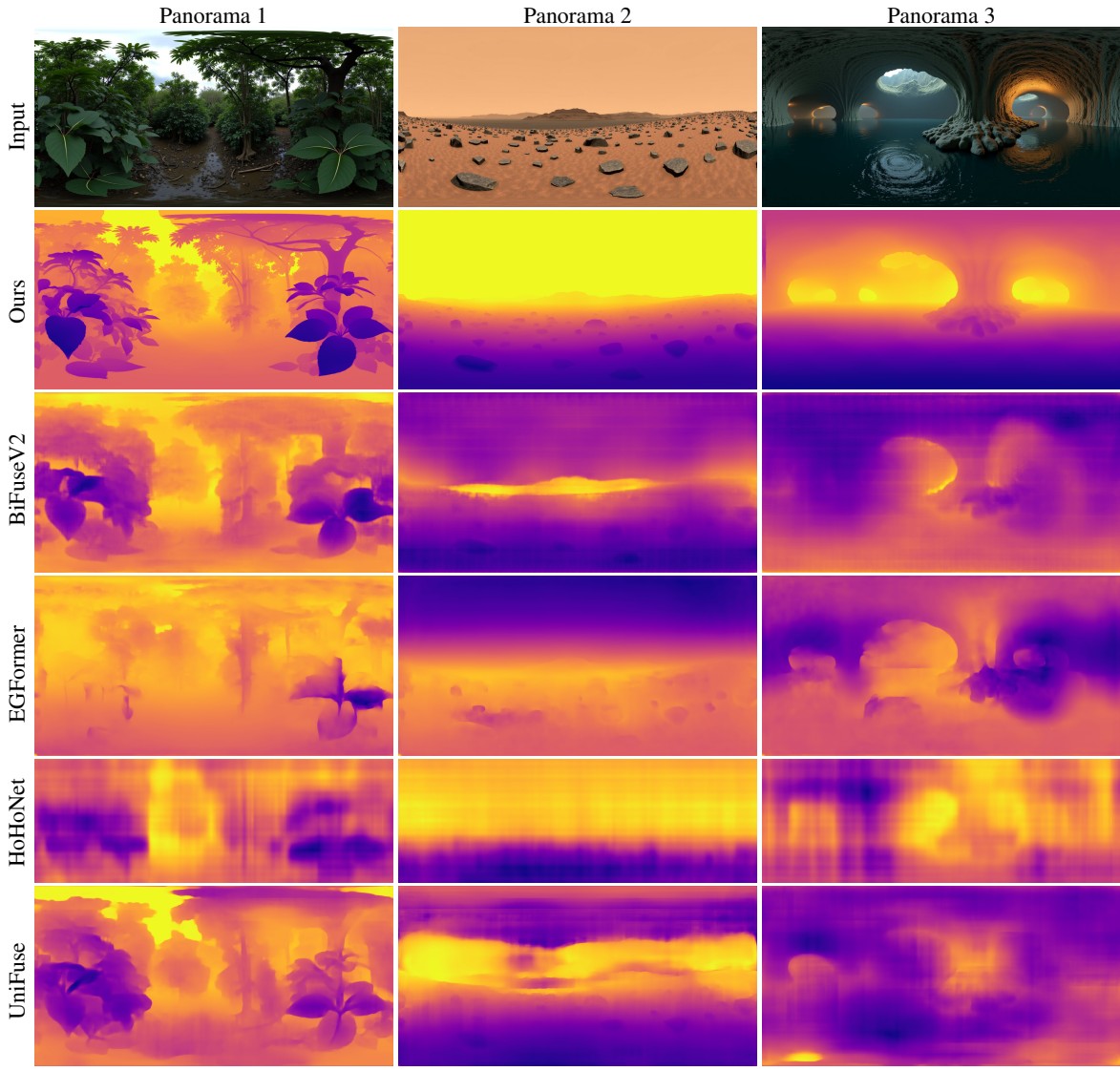

*Figure 11.* **Qualitative comparison of depth predictions.** Top row: input panorama. Subsequent rows: depth predictions for Spherical-Dreamer (via 360MonoDepth (Rey–Area et al., 2022)), BiFuseV2 (Wang et al., 2023a), EGFormer (Yun et al., 2023), HoHoNet (Sun et al., 2021) and UniFuse (Jiang et al., 2021). Our depth maps are the most accurate with the fewest artifacts.

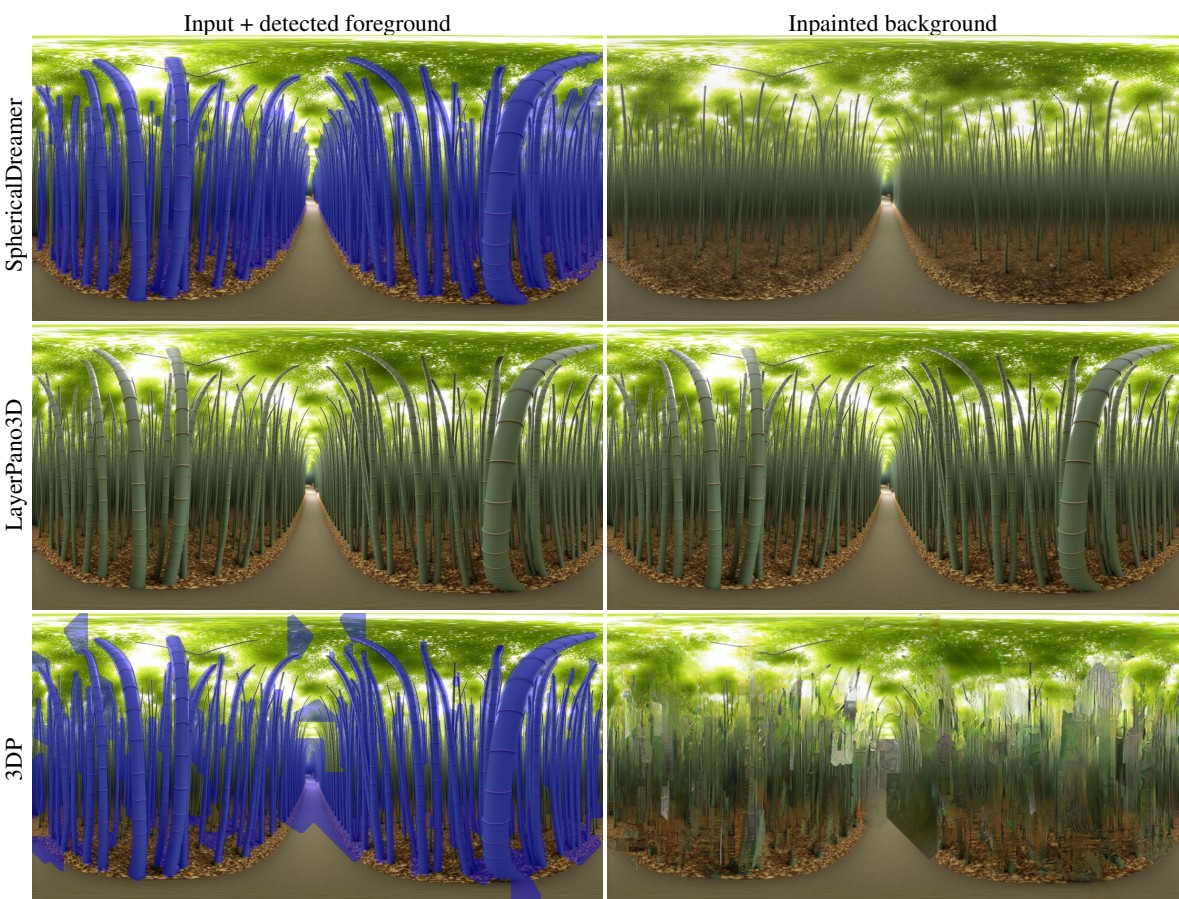

*Figure 12.* **LDP qualitative comparison.** Each row shows, for one method, the input panorama with detected foreground (left) and the inpainted background after foreground removal (right).

Input + detected foreground          Inpainted background

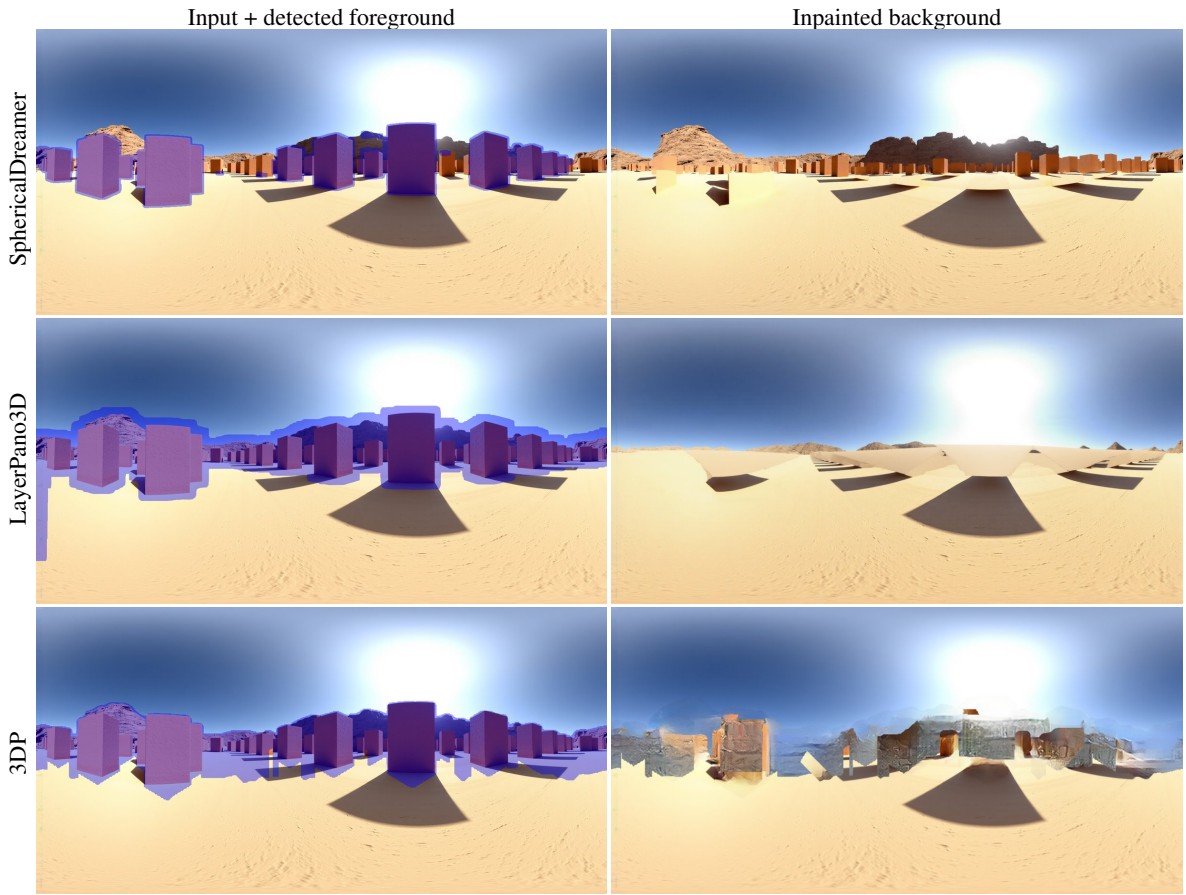

*Figure 13.* **LDP qualitative comparison.** Each row shows, for one method, the input panorama with detected foreground (left) and the inpainted background after foreground removal (right).

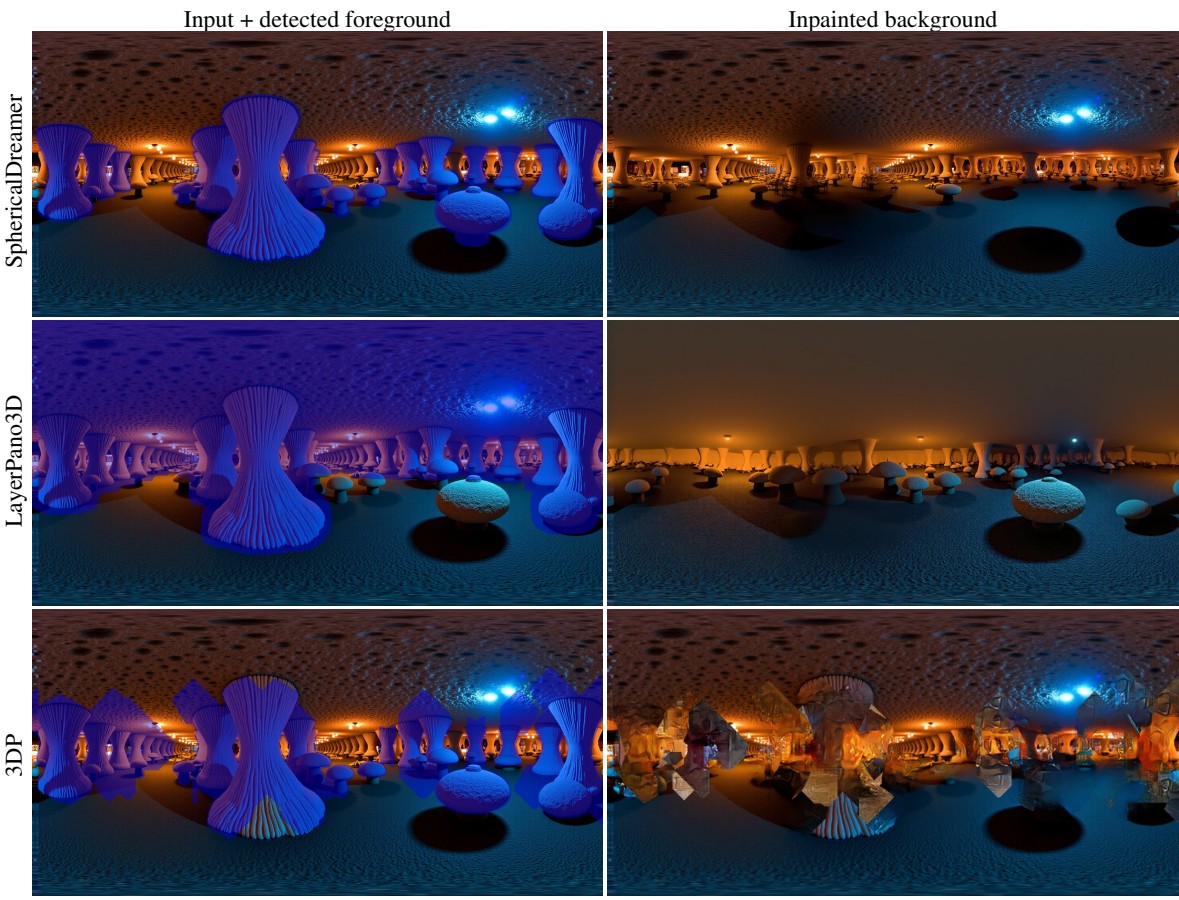

*Figure 14.* **LDP qualitative comparison.** Each row shows, for one method, the input panorama with detected foreground (left) and the inpainted background after foreground removal (right).

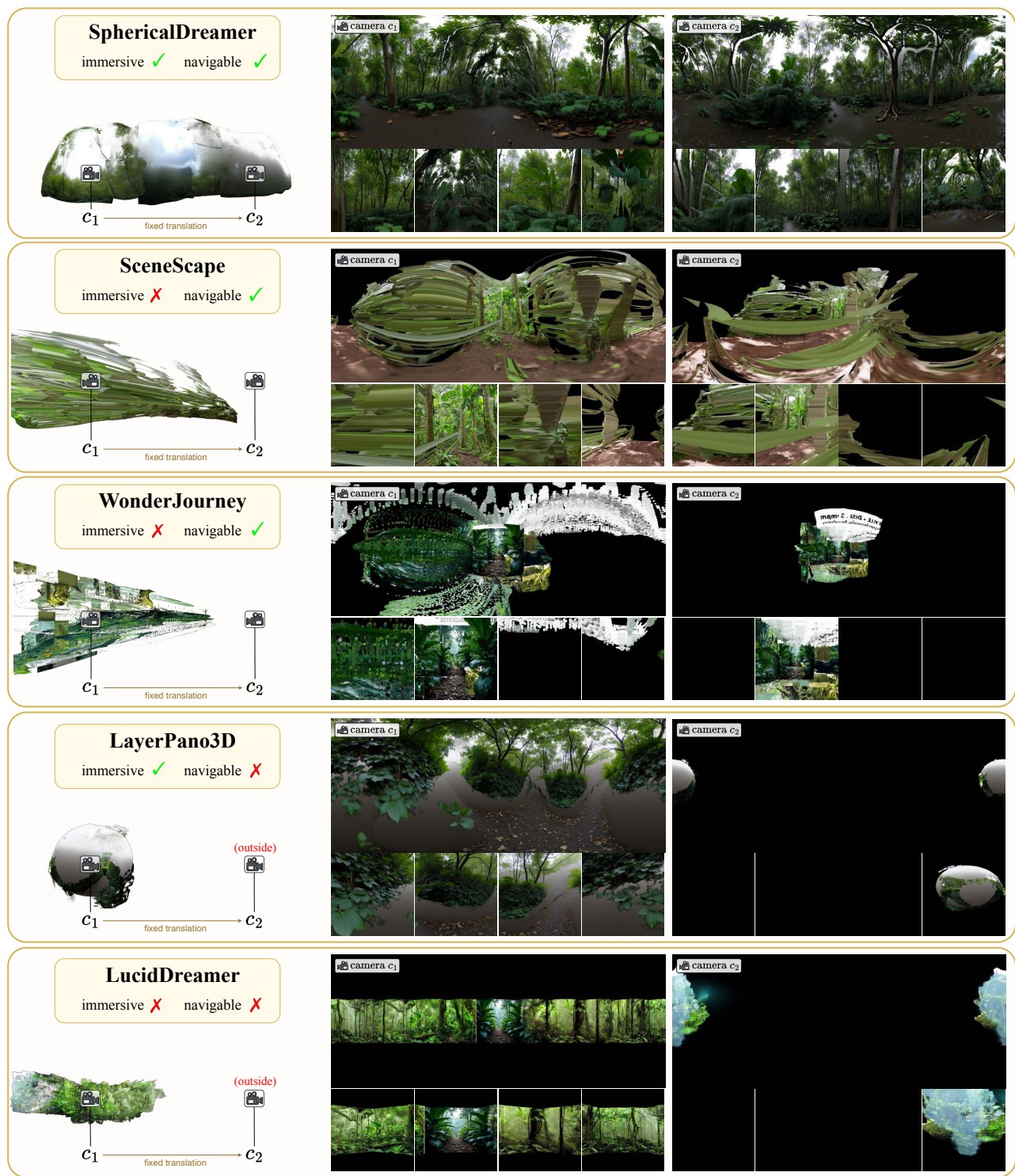

*Figure 15.* **Qualitative comparison over the full** $180° \times 360°$ **field of view across distant camera viewpoints.** Additionnal results on *a jungle*. Perspective renderings (left, front, right, back) are included to complement each panoramic rendering.

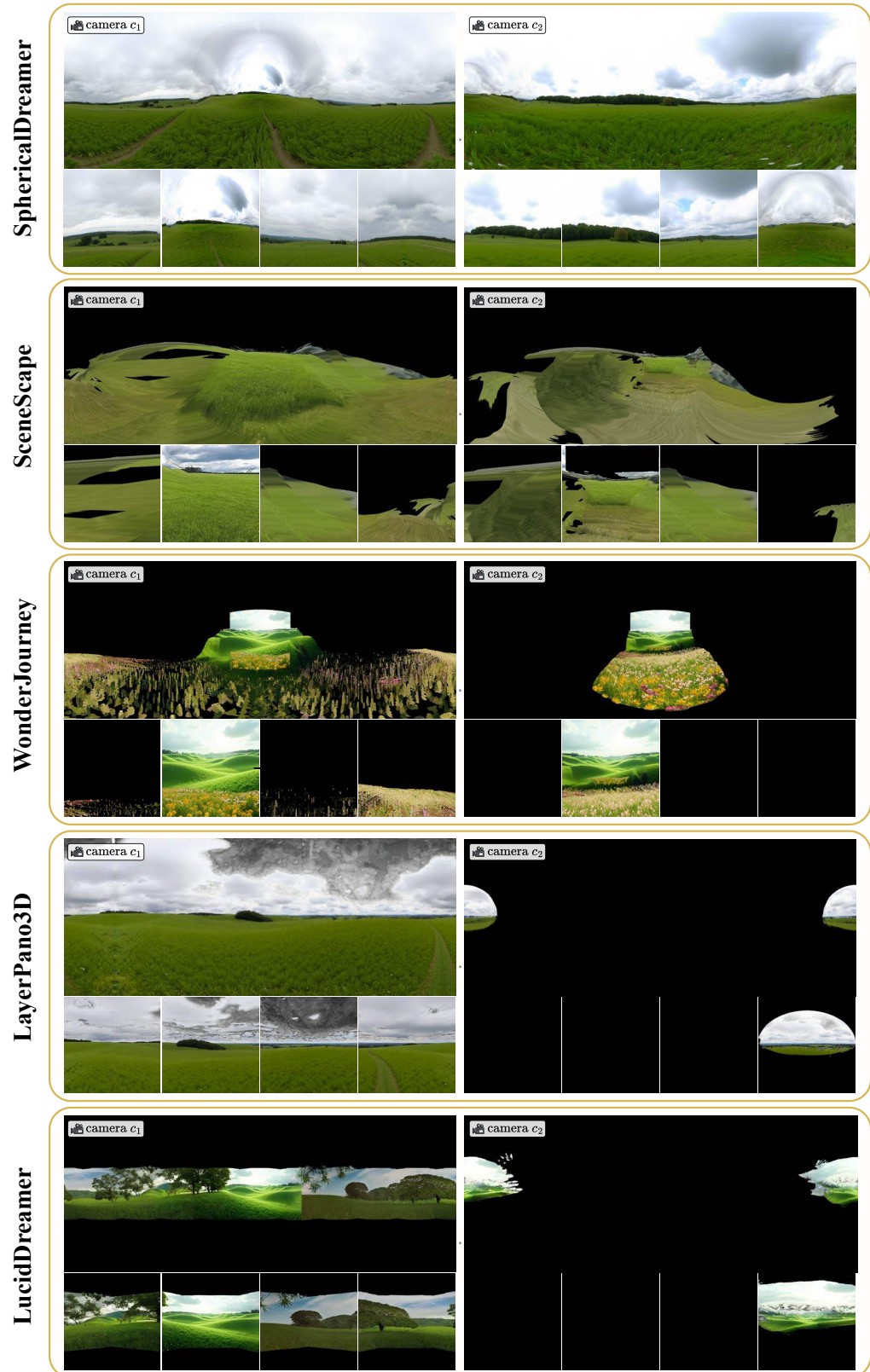

*Figure 16.* **Qualitative comparison over the full** $180° \times 360°$ **field of view across distant camera viewpoints.** Additionnal results on *a grass field*. Perspective renderings (left, front, right, back) are included to complement each panoramic rendering.

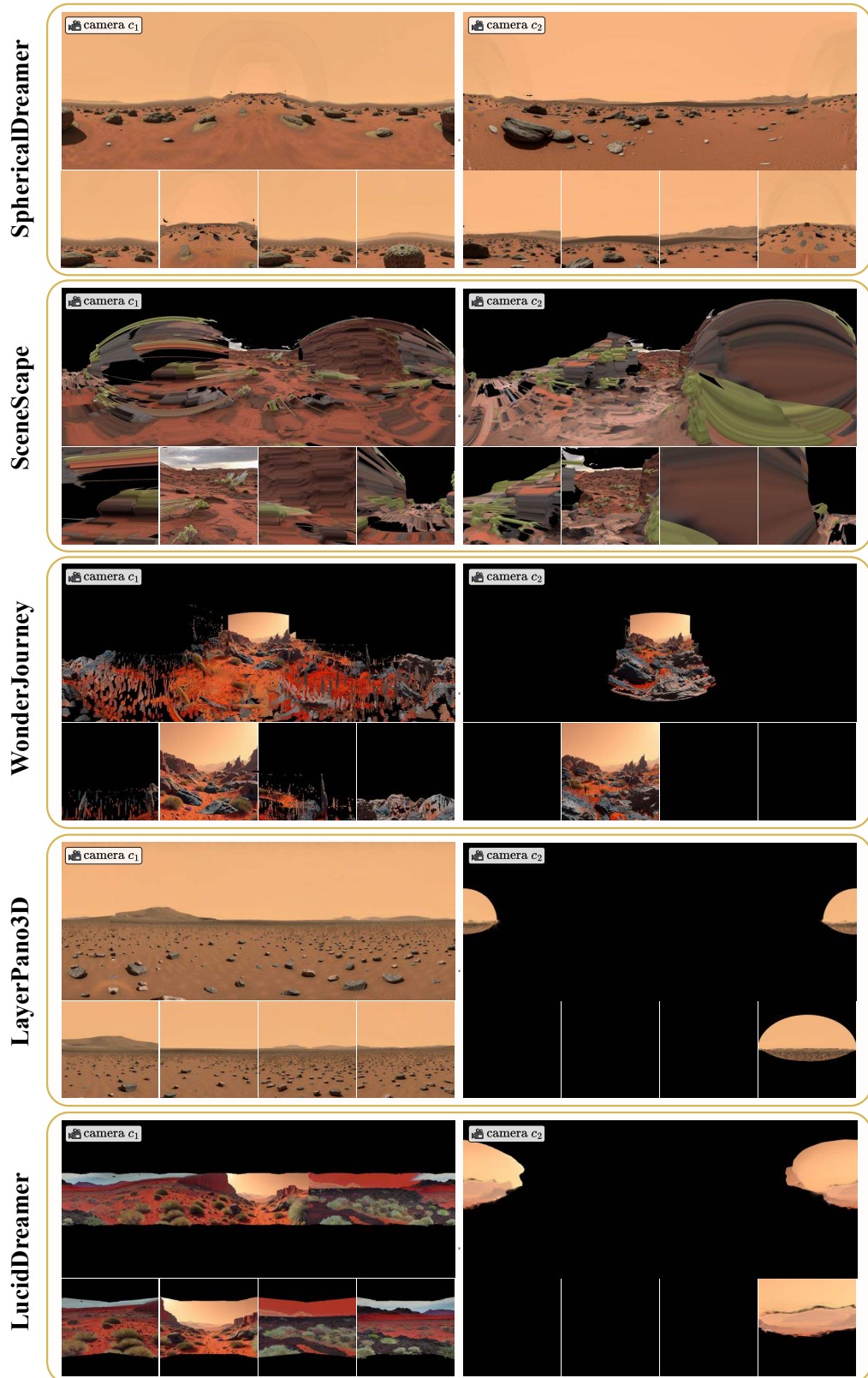

*Figure 17.* **Qualitative comparison over the full** $180° \times 360°$ **field of view across distant camera viewpoints.** Additionnal results on *a martian desert*. Perspective renderings (left, front, right, back) are included to complement each panoramic rendering.

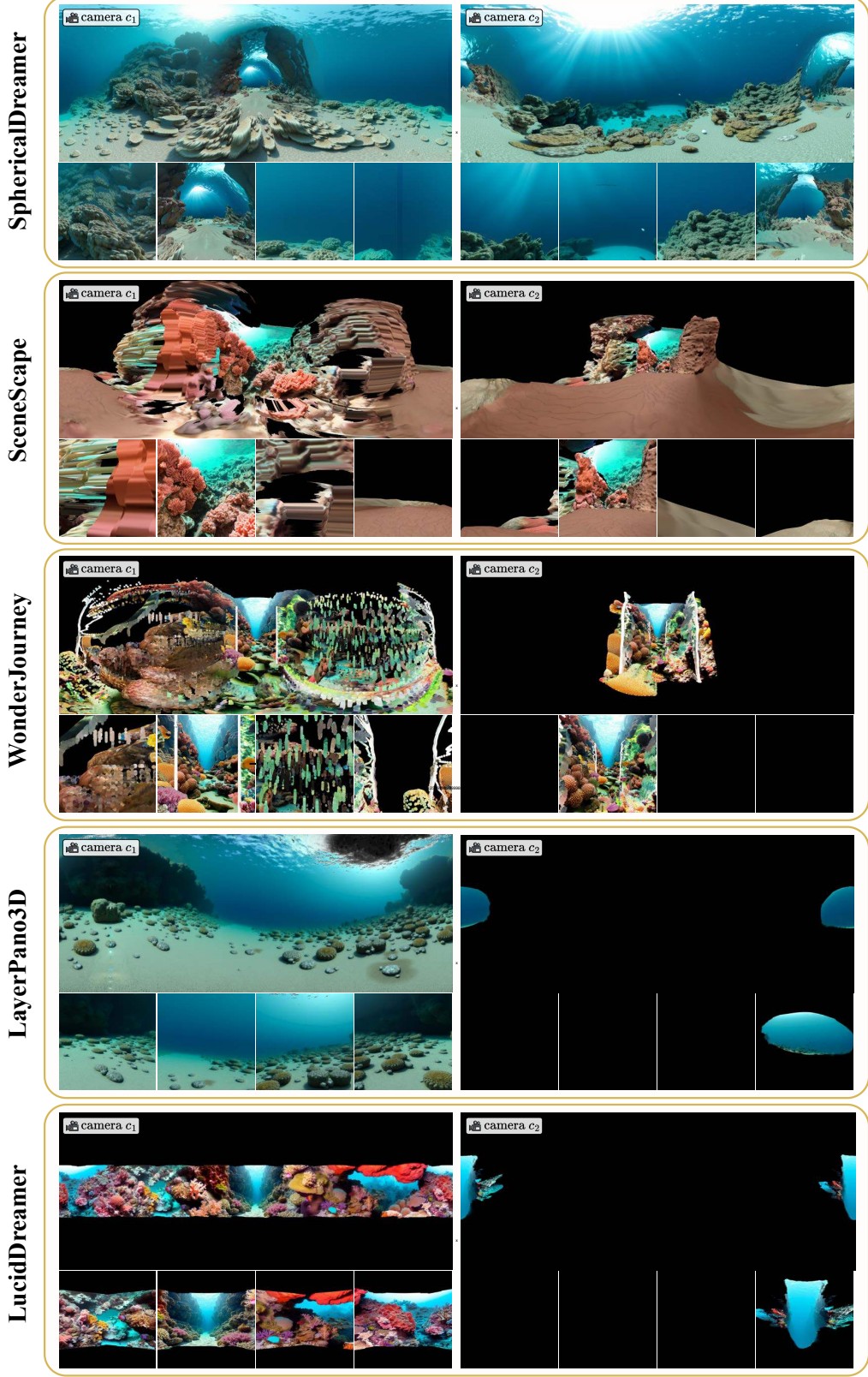

*Figure 18.* **Qualitative comparison over the full** $180° \times 360°$ **field of view across distant camera viewpoints.** Additionnal results on *an underwater world*. Perspective renderings (left, front, right, back) are included to complement each panoramic rendering.

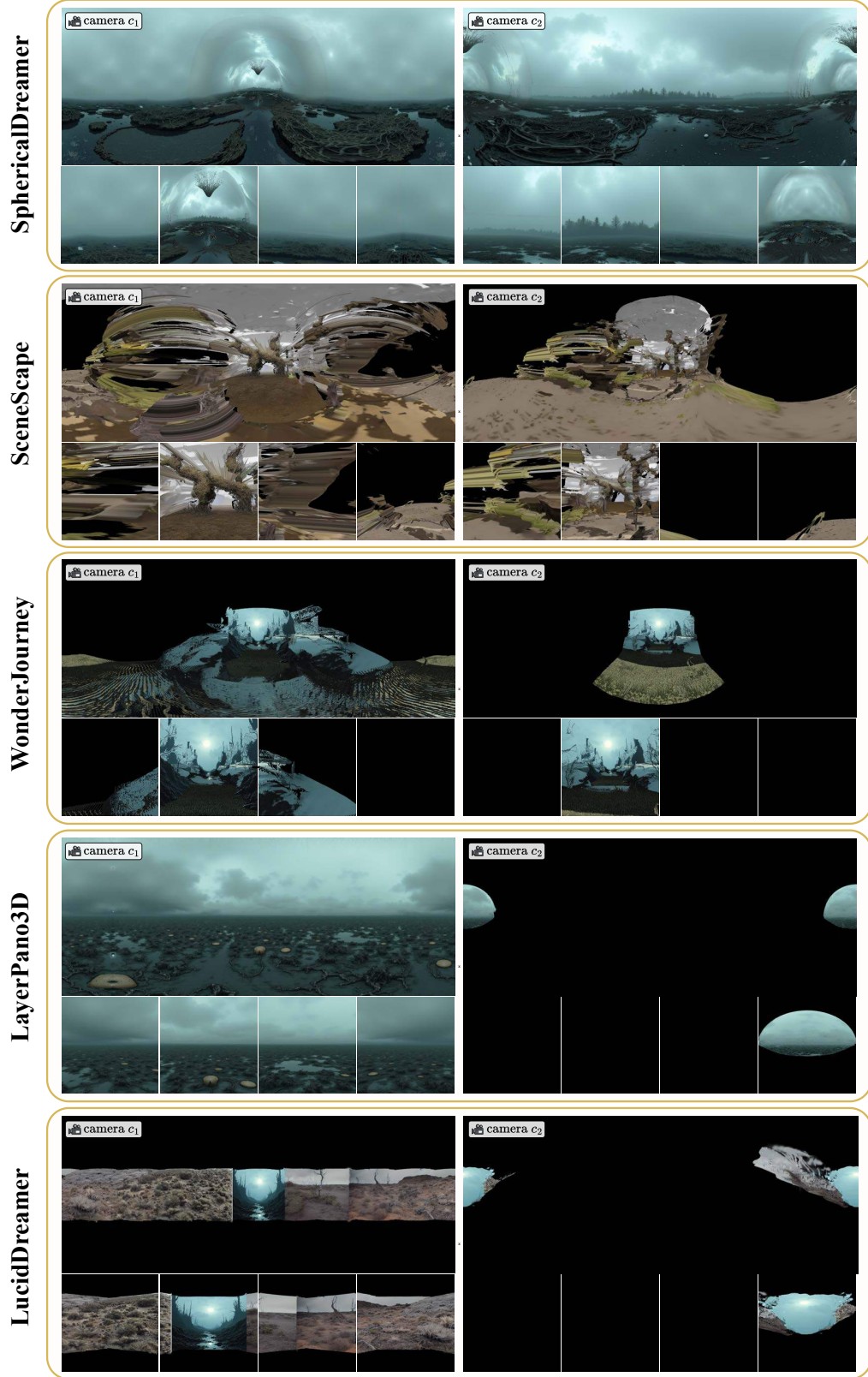

*Figure 19.* **Qualitative comparison over the full** $180° \times 360°$ **field of view across distant camera viewpoints.** Additionnal results on *a desolated landscape*. Perspective renderings (left, front, right, back) are included to complement each panoramic rendering.

