# OpenReview forum: "SphericalDreamer: Generating Navigable Immersive 3D Worlds with Panorama Fusion"
_ICML.cc/2026/Conference — ICML 2026 regular_

### Official Review · Reviewer_TrMv · 2026-03-07

**Soundness:** 2
**Presentation:** 3
**Significance:** 3
**Originality:** 3
**Overall Recommendation:** 5
**Confidence:** 4

**Summary:**

The paper introduces SphericalDreamer, a framework designed to generate 3D environments from text prompts that are simultaneously navigable over long distances and fully immersive ($360^\circ \times 180^\circ$ field of view). The method first generates multiple layered depth panoramas (LDPs) to handle occlusion artifacts and lifts them into 3D point clouds, which act as building blocks. To connect these spherical blocks, the authors render intermediate views, inpaint missing regions using FluxFill, and propose a novel "Harmonic Blending" technique. This blending method uses Laplacian energy minimization to smoothly align newly estimated depth maps with existing reliable geometry, eliminating visible seams. Finally, all blocks are assembled into a unified 3D world.

**Compliance With Llm Reviewing Policy:**

Affirmed.

**Final Justification:**

The authors have addressed the core weakness of insufficient evaluation by adding multiple standard metrics (CLIP-Score, C-CLIP, CLIP-IQA, Q-Align, geometric metrics) and informative ablation studies. The Harmonic Blending technique remains a solid and original contribution, and the expanded experiments now provide adequate evidence for the claimed advantages. While a user study and more diverse trajectory evaluation would further strengthen the work, the current revision meets the bar for acceptance. We raise our score and confidence.

**Key Questions For Authors:**

1. Why were standard generative evaluation metrics omitted? Please provide quantitative results for text-alignment (e.g., CLIP-Score) and geometric consistency (e.g., Reprojection Error or SI-RMSE).

2. The current evaluation primarily focuses on straight-line horizontal displacements. How robust is the Harmonic Blending technique when dealing with complex, highly non-linear, or vertical camera trajectories?

**Limitations:**

yes

**Strengths And Weaknesses:**

**Strengths**

1. The theoretical foundation for seamlessly fusing geometric structures is solid. The "Harmonic Blending" technique creatively adapts classical Laplacian surface editing for depth map deformation. This mathematical formulation ensures smooth transitions without geometric discontinuities.
2. The paper is clearly structured and well-written. The qualitative visual comparisons against baselines like SceneScape, WonderJourney, and LucidDreamer effectively illustrate the proposed method's superiority in maintaining both immersivity and navigability.
3. While the pipeline relies heavily on existing off-the-shelf 2D diffusion models and depth estimators , the specific combination of Layered Depth Panoramas (LDPs) and graph-based harmonic blending offers a highly original and effective solution to the multi-panorama fusion problem.

**Weaknesses**

1. The experimental evaluation is severely inadequate for a top-tier venue. The authors rely exclusively on BRISQUE for image quality and a custom "Coverage" metric for immersivity. There is a complete absence of text-alignment metrics (e.g., CLIP-Score) to verify prompt adherence, geometric consistency metrics (e.g., Reprojection Error, SI-RMSE), and human evaluation (User Study), which are standard in evaluating discrete generative models and 3D scenes.
2. The quantitative evaluation section fails to properly contextualize the results against standard community benchmarks, making the claims of state-of-the-art visual fidelity difficult to verify objectively.

---

> ### Author Rebuttal · Authors · 2026-03-30
>
> We thank the reviewer for recognizing the **solid theoretical foundation**, the method's **superiority in maintaining immersivity and navigability**, and the **highly original and effective solution** to multi-panorama fusion.
>
> All referenced tables, figures, and animated results are available at: https://anon-supp.github.io/
>
> ## Extended evaluation
>
> In this rebuttal, we strengthen the evaluation with (i) **new experiments** to assess the visual quality, consistency, and geometric accuracy using new metrics, (ii) **new ablations**, and (iii) **new detailed analysis via experiments** on our core submodules (LDI and Harmonic Blending).
>
> ### Quality evaluation
>
> We conduct **new experiments** where we utilize CLIP-IQA and Q-Align alongside BRISQUE. SphericalDreamer achieves the best quality on all three metrics under combined rotation and translation (Section G, Table 6).
>
> ### Text-alignment and semantic consistency
>
> We conduct **new experiments** where we evaluate text alignment using via CLIP-Score and global semantic consistency via C-CLIP. Our method achieves the **best results** under both rotation and translation, confirming prompt alignment and high global semantic consistency across the generated worlds (Section G, Table 5).
>
> ### Geometry evaluation
>
> We evaluate geometry via two complementary approaches.
>
> #### a. Evaluation via predicted depth
>
> We provide **new comparisons** between depth maps predicted by SphericalDreamer and those from other 360° depth estimation models.
>
> **Qualitatively** (Section D, Figure 4), our depth maps are the most accurate with the fewest artifacts.
>
> **Quantitatively** (Section D, Table 3), we evaluate on Replica2K using AbsRel, RMSE, SI-RMSE, and accuracy thresholds (δ<1.25, δ<1.25², δ<1.25³). Our approach **consistently outperforms all baselines** across every metric.
>
> #### b. Evaluation via image quality metrics
>
> The image quality metrics (CLIP-IQA, Q-Align) presented above also reflect geometric consistency, as geometric artifacts degrade visual quality. Our best-in-class results further support the quality of the underlying geometry.
>
> ### New ablation study
>
> We ablate two key components: Harmonic Blending (HB) and LDI. We remove LDI, and replace HB with naïve blending, bilinear interpolation, and InFusion [5] (diffusion-based depth inpainting).
>
> **Qualitatively** (Section B, Figure 2), replacing HB causes geometry artifacts in transition regions; removing LDI introduces occlusion artifacts with unfilled regions. The full model produces complete, coherent environments.
>
> **Quantitatively** (Section B, Table 2), we evaluate C-CLIP, CLIP-Score, BRISQUE, CLIP-IQA, Q-Align, and Coverage. Both ablations cause consistent drops across all metrics. HB improves completeness and visual quality; LDI is critical for handling occlusions and maintaining consistency.
>
> ### Further evaluation of HB and LDI
>
> #### LDI comparison with prior approaches
>
> We provide **new comparisons** of our LDI with LayerPano3D and 3D Photography in Section C, Figure 3. SphericalDreamer produces more realistic background panoramas without artifacts, owing to accurate foreground segmentation and occlusion-aware mask estimation.
>
> #### Harmonic blending vs. simpler alternatives
>
> We provide a **new comparison** of harmonic blending with simpler approaches for the task of depth completion. Specifically, we mask regions of reference depth maps and reconstruct them with each method.
>
> **Qualitatively** (Section E, Figure 5), HB produces the smoothest transitions without artifacts.
>
> **Quantitatively** (Section E, Table 4), HB achieves the best **Depth Transition Score** (boundary discontinuity; lower = smoother) and **Depth Estimation Error** (MAE vs. ground-truth near boundary), confirming seamless alignment in overlapping areas.
>
> We will include this analysis in the final paper.
>
> ## Dealing with highly non-linear camera trajectories
>
> Harmonic blending is formulated in a general way and is not restricted to linear trajectories. For instance, Figure 8 of the paper illustrates its application to highly non-linear trajectories. However, for simplicity and to enable direct comparison with prior work, we focus on linear trajectories in our experiments.
>
> ---
>
> We sincerely hope that our response resolves the concerns raised. If the reviewer has other technical points to discuss that may currently prevent them from reconsidering the score, we would be more than happy to discuss further.
>
> ## References
>
> [1] Wang, F.-E., et al. (2023). BiFuse++. IEEE TPAMI, 45(5), 5448–5460.
> [2] Yun, I., et al. (2023). EGformer. ICCV, 6078–6089.
> [3] Sun, C., Sun, M., & Chen, H.-T. (2021). HoHoNet. CVPR, 2573–2582.
> [4] Su, D., et al. (2025). UniFuse. ICCV, 14238–14247.
> [5] Liu, Z., et al. (2024). InFusion. arXiv:2404.11613.

---

> > ### Author Rebuttal · Reviewer_TrMv · 2026-04-02
> >
> > We thank the authors for the comprehensive rebuttal. The addition of CLIP-Score, C-CLIP, geometric metrics on Replica2K, and the new ablation studies substantially address our main evaluation concerns. The clarification regarding non-linear trajectories via Figure 8 is reasonable, though systematic quantitative evaluation on diverse trajectory types would further strengthen the paper.

---

### Official Review · Reviewer_dn3a · 2026-03-11

**Soundness:** 2
**Presentation:** 3
**Significance:** 3
**Originality:** 2
**Overall Recommendation:** 4
**Confidence:** 4

**Summary:**

This paper aims to enable the generation of immersive (panoramic) and navigable (different camera views) 3D environments, conditioned on text inputs. Specifically, the authors propose SphericalDreamer, consisting of Spherical Building Blocks Generation, Pairwise Generative Fusion, and World Assembly. Experiments are conducted over three types of camera trajectories and achieve better performance than baselines.

**Compliance With Llm Reviewing Policy:**

Affirmed.

**Final Justification:**

The authors resolved my concern well in the rebuttal.

**Key Questions For Authors:**

1. Can the authors add some cross-view consistency metrics and compare with baselines? Like using CLIP-based consistency (C-CLIP) and style consistency (C-Style) in PanoDreamer, but computed on overlapping or corresponding regions rendered from different views? This would help quantify whether the generated world remains consistent across translation, rather than only measuring single-view quality.


2. Can the authors provide an ablation studyfor the proposed Layer Depth Panorama, comparing it with other choices such as single-depth panorama and the four-layer LDI representation in PanoDreamer? This would help justify the module design.


3. Can the authors report performance as the world size N increases? Since the current experiments are fixed to N=3, it is difficult to evaluate its generalization when the number increases.

**Limitations:**

yes

**Strengths And Weaknesses:**

### Strengths

1. The topic of generating immersive and navigable 3D environments is interesting, and the proposed three-step method is intuitive and clear.

2. The paper is well written and well structured, making it easy to follow.

3. Experiments are conducted, showing the effectiveness of the proposed method.

### Weaknesses

1. **Limited evaluation.**
The current evaluation mainly focuses on quality and coverage. However, the alignment across views, like the geometry consistency, is not evaluated, making it hard to have a comprehensive understanding.

2. **Additional ablation studies and analysis are needed.**
The paper proposes some new modules, like Layer Depth Panorama, but they are not evaluated and compared with other methods.

---

> ### Author Rebuttal · Authors · 2026-03-30
>
> We thank the reviewer for finding our work **intuitive and clear**, **well structured**, with **experiments showing the effectiveness of our method**.
>
> All referenced tables, figures, and animated results are available at: https://anon-supp.github.io/
>
> ## (w.1 + q1) Evaluation of geometry, cross-view consistency, and quality
>
> In this rebuttal, we introduce **new** quantitative and qualitative analyses assessing geometric fidelity, cross-view consistency, and perceptual quality.
>
> ### (q1) Cross-view consistency
>
> We conduct **new experiments** where we compute CLIP-Score and C-CLIP between various views of generated worlds. These metrics measure view consistency w.r.t. text (CLIP-Score) and other views (C-CLIP). Our method achieves the best results under both rotation and translation, confirming semantic consistency over large distances (Section G, Table 5).
>
> We also confirm this **qualitatively** by generating larger worlds (Section A, Figure 1).
>
> ### (w1) Geometry
>
> We evaluate geometry via two complementary approaches: direct depth assessment and indirect perceptual quality metrics.
>
> #### (w1.a) Evaluation via predicted depth
>
> We provide **new comparisons** between the depth maps predicted by SphericalDreamer and those from other 360° depth estimation models.
>
> **Qualitatively** (Section D, Figure 4), our depth maps are the most accurate with the fewest artifacts.
>
> **Quantitatively** (Section D, Table 3), we evaluate on Replica2K using AbsRel, RMSE, SI-RMSE, and accuracy thresholds (δ<1.25, δ<1.25², δ<1.25³). Our approach **consistently outperforms all baselines** across every metric.
>
> #### (w1.b) Evaluation via image quality metrics
>
> We assess geometry indirectly using CLIP-IQA and Q-Align, which implicitly measure the quality of the underlying geometry, as geometry artifacts degrade visual quality. Our method achieves the best results under combined rotation and translation, validating underlying geometry quality (Section G, Table 6).
>
> ## (w.2 + q.2) Ablation studies and analysis
>
> ### Ablation of LDI
>
> We propose **new ablations** of LDI and compare with the full pipeline.
>
> **Qualitatively** (Section B, Figure 2), removing LDI causes visible occlusion artifacts with unfilled background regions.
>
> **Quantitatively** (Section B, Table 2), quality, coverage, and cross-view consistency metrics all drop without LDI.
>
> ### Comparison of SphericalDreamer's LDI with prior approaches
>
> We provide **new comparisons** of our LDI construction with LayerPano3D and 3D Photography in Section C, Figure 3. SphericalDreamer produces more realistic background panoramas without artifacts, owing to more accurate foreground segmentation and occlusion-aware mask estimation.
>
> ## (q.3) Bigger world sizes
>
> We provide **new experiments** with worlds of varying size (N=3 to N=7).
>
> **Qualitatively** (Section A, Figure 1), SphericalDreamer produces short to long-range environments.
>
> **Quantitatively** (Section A, Table 1), BRISQUE, Coverage, CLIP-Score, C-CLIP, CLIP-IQA, and Q-Align remain stable as N increases, confirming maintained quality and consistency.
>
> ---
>
> We sincerely hope that our response resolves the concerns raised. If the reviewer has other technical points to discuss that may currently prevent them from reconsidering the score, we would be more than happy to discuss further.
>
> ## References
>
> [1] Wang, F.-E., et al. (2023). BiFuse++. IEEE TPAMI, 45(5), 5448–5460.
>
> [2] Yun, I., et al. (2023). EGformer. ICCV, 6078–6089.
>
> [3] Sun, C., Sun, M., & Chen, H.-T. (2021). HoHoNet. CVPR, 2573–2582.
>
> [4] Su, D., et al. (2025). UniFuse. ICCV, 14238–14247.

---

> > ### Author Rebuttal · Reviewer_dn3a · 2026-04-02
> >
> > The response addressed my concerns well. Good luck.

---

### Official Review · Reviewer_Xn4Z · 2026-03-12

**Soundness:** 3
**Presentation:** 3
**Significance:** 2
**Originality:** 3
**Overall Recommendation:** 4
**Confidence:** 4

**Summary:**

The paper introduces SphericalDreamer, a method for generating fully immersive, long-range 3D environments from textual prompts. The proposed approach enables high-quality scene synthesis with full omnidirectional coverage, supporting consistent rendering across distant and wide-ranging camera viewpoints.

**Compliance With Llm Reviewing Policy:**

Affirmed.

**Final Justification:**

Thanks for the rebuttal. My concerns have been addressed.

**Key Questions For Authors:**

The paper would benefit from ablation studies investigating the effect of varying the number of blocks and panoramas. In particular, it would be interesting to understand the maximum number of panoramas that can be assembled while still maintaining high geometric and texture consistency.

The manuscript lacks sufficient analysis of training cost and inference efficiency. Additional experiments comparing runtime performance would be valuable—for example, reporting the time required for a single scene generation and analyzing how the runtime scales as the number of panoramas increases.

**Limitations:**

yes

**Strengths And Weaknesses:**

Strengths

The paper presents an interesting approach to addressing limitations in existing panoramic scene generation methods, enabling long-range panoramic synthesis with more flexible viewpoint control, which is particularly impressive. The experimental results also demonstrate strong performance, suggesting the effectiveness of the proposed framework. I think the proposed idea is both novel and promising.

Weaknesses

Despite these strengths, several issues require further clarification and investigation:

1. The method relies on a pretrained monocular depth estimator for panoramic images, which may introduce estimation errors, particularly for generated images. Additional analysis of the robustness of the method to depth estimation errors, along with comparative experiments using different depth estimators or perturbations, would strengthen the evaluation.

2. The generalization ability of the proposed approach is not sufficiently demonstrated. It would be helpful to provide results on more diverse scene categories, such as urban environments, driving or road scenes, and indoor environments, to better assess the method’s applicability across different domains.

3. During the world assembly stage, there may be concerns regarding alignment errors in overlapping regions. Simply stitching adjacent spheres and filling blocks could lead to geometric distortions or occlusion inconsistencies in overlapping areas. The authors are encouraged to provide a more detailed analysis of these potential limitations, along with additional visualizations illustrating such cases.

---

> ### Author Rebuttal · Authors · 2026-03-30
>
> We thank the reviewer for finding our work **particularly impressive**, with **strong performance** and an idea that is **both novel and promising**.
>
> All referenced tables, figures, and animated results are available at: https://anon-supp.github.io/
>
> ## (w.1) On the reliance on monocular depth estimators
>
> While inaccuracies in monocular depth estimation degrade geometric quality, this limitation is **not specific to SphericalDreamer**: all compared baselines equally rely on depth estimation. This is a **broader limitation of the current paradigm**, not of our method. We will include additional discussion in the final paper.
>
> ## (w.2) More diverse scenes
>
> As noted in our limitations, SphericalDreamer is better suited to outdoor/natural environments due to its reliance on spherical imagery, which is less effective for structured planar geometry (urban/indoor scenes). In the camera-ready, we will explicitly scope our claims to outdoor and natural settings in the abstract, introduction, and experiments, preventing misinterpretation of the method as universally applicable.
>
> ## (w.3) Alignment of adjacent spheres
>
> The **only potential source of geometric misalignment** is at the panorama fusion stage. World assembly is a direct aggregation of already-aligned blocks and introduces no additional distortion. **Harmonic blending** ensures smooth geometric transitions at fusion boundaries.
>
> We validate this through **two new evaluations for this rebuttal**:
>
> ### Ablation within our pipeline
>
> We replace Harmonic Blending (HB) with naïve blending, bilinear interpolation, and InFusion [1] (diffusion-based depth inpainting).
>
> **Qualitatively** (Section B, Figure 2), HB outperforms all alternatives, fully completing all missing regions.
>
> **Quantitatively** (Section B, Table 2), we compute C-CLIP, CLIP-Score, BRISQUE, CLIP-IQA, and Q-Align. HB outperforms all alternatives on every metric.
>
> ### In-depth analysis
>
> We mask portions of reference depth maps and reconstruct them with each method.
>
> **Qualitatively** (Section E, Figure 5), HB provides the smoothest transitions without artifacts.
>
> **Quantitatively** (Section E, Table 4), we report **Depth Transition Score** (boundary discontinuity; lower = smoother) and **Depth Estimation Error** (MAE vs. ground-truth near boundary). HB achieves the best scores on both, confirming that panorama fusion **does not exhibit alignment issues** at overlapping regions.
>
> We will include this discussion in the final paper.
>
> ## (q.1) Varying the number of building blocks
>
> We provide **new experiments** with worlds of varying size (N=3 to N=7).
>
> **Qualitatively** (Section A, Figure 1), SphericalDreamer produces short to long-range environments.
>
> **Quantitatively** (Section A, Table 1), BRISQUE, Coverage, CLIP-Score, C-CLIP, CLIP-IQA, and Q-Align remain stable as N increases, confirming maintained quality and consistency. No intrinsic limit on N was observed; the main constraint is **computational cost**.
>
> ## (q.2) Runtime performance
>
> We report in a **new table** the total runtime and per-component breakdown for N=2, N=3, and N=4 on a single NVIDIA A100 GPU (Section H, Table 7).
>
> Generating a 3D world with N=3 panoramas and rendering a video trajectory takes ~50 minutes. Generation time scales approximately linearly with N.
>
> ---
>
> We sincerely hope that our response resolves the concerns raised. If the reviewer has other technical points to discuss that may currently prevent them from reconsidering the score, we would be more than happy to discuss further.
>
> ## References
>
> [1] Liu, Z., Ouyang, H., Wang, Q., Cheng, K. L., Xiao, J., Zhu, K., … Cao, Y. (2024). InFusion: Inpainting 3D Gaussians via Learning Depth Completion from Diffusion Prior. arXiv:2404.11613.

---

> > ### Author Rebuttal · Reviewer_Xn4Z · 2026-04-03
> >
> > Thanks for the rebuttal; my concerns have been addressed. I encourage the authors to release the code and model weights.

---

> > > ### Author Response · Authors · 2026-04-03
> > >
> > > The code and model weights will indeed be released upon publication.
> > >
> > > We are glad we have resolved all of the reviewer's concerns. In light of this, we kindly invite the reviewer to reconsider their score.
> > >
> > > We thank the reviewer again for all of the constructive feedback.

---

### Official Review · Reviewer_dnTD · 2026-03-13

**Soundness:** 3
**Presentation:** 3
**Significance:** 4
**Originality:** 3
**Overall Recommendation:** 3
**Confidence:** 5

**Summary:**

This paper proposes SphericalDreamer, a pipeline for text-to-3D world generation that tries to combine two goals that are usually separated in prior work: full 360-degree immersive viewing and longer-range navigation. The method generates multiple panoramas from text, lifts each panorama into a layered point-cloud "sphere," opens adjacent spheres, renders an intermediate view between them, inpaints the missing region, estimates depth, and uses a harmonic blending step to align new depth with existing geometry before back-projecting the filled content into 3D.
The problem is important, and the paper presents an intuitive system with appealing qualitative results on several natural outdoor scenes. However, the claimed contribution is only partly supported. Much of the method is a combination of existing components, and the new technical piece, harmonic blending, appears closer to a standard Laplacian or harmonic deformation idea than a strong new learning contribution. More importantly, the paper does not convincingly show that semantics and geometry remain globally consistent as the viewer moves through the world (e.g., the straight tree in the video demo). The experiments are also too narrow to support the strong "first" and "state-of-the-art" claims.

**Compliance With Llm Reviewing Policy:**

Affirmed.

**Key Questions For Authors:**

The method is largely a system built from existing parts: text-to-panorama generation, SAM-based masking, inpainting, monocular depth estimation, point-cloud lifting, and a classical smooth deformation style blending step. The "harmonic blending" formulation does not appear fundamentally new, and the paper does not provide strong evidence that this component is necessary for the overall gains. There is no meaningful quantitative ablation against simpler alternatives beyond a small visual comparison.


Adjacent spheres are generated largely independently and then connected with local image inpainting plus local depth alignment. That can smooth boundaries, but it does not guarantee persistent world consistency (3d geometry with semantic consistency) when the agent moves through the environment. There is no explicit global 3D consistency objective, no object correspondence across panoramas, no shared scene memory, no loop closure, and no global optimization over all generated views. Because of that, it is unclear why object identity, scale, layout, and semantics should remain stable beyond local transitions. Addressing viewpoint-consistent semantics and geometry would likely require explicit cross-view or 3D-global constraints, such as multi-view consistency losses, object-level correspondences, scene graph or world-state memory, or global refinement over all panoramas rather than pairwise local fusion.

The evaluation uses only a small set of prompts, mostly natural outdoor scenes, with a fixed straight-line setup and only N = 3 panoramas. That is not enough to establish "long-range" (typically for many meters) navigation in a broad sense.

**Limitations:**

This is an interesting and nicely presented system paper with strong demos, but I am not convinced it makes a sufficiently strong ML contribution in its current form. The central limitation is that the paper claims immersive navigation in a coherent generated world, yet the method only enforces local geometric stitching, not global semantic or structural consistency. The experiments do not directly test that core claim, and the evaluation scope is too narrow for the very strong novelty and performance statements. I would be more positive if the paper added stronger ablations, direct consistency metrics, broader scene types, and a clearer argument or mechanism for maintaining persistent semantics and geometry under viewpoint changes.

**Strengths And Weaknesses:**

The paper targets a real gap in current 3D world generation: panorama-based methods are immersive but spatially limited, while iterative expansion methods can move farther but often lose full view coverage. Framing the task around both immersion and navigation is useful.

The system is easy to follow. The layered depth panorama idea, sphere opening, midpoint rendering, and pairwise fusion form a coherent pipeline. The qualitative examples are visually appealing, especially for natural scenes such as jungles, caves, and grass fields.

The harmonic blending step is sensible as a way to reduce seams between trusted rendered depth and newly estimated depth. Even if the idea is not completely novel, it is a practical design choice, and the appendix gives enough detail to understand the intended behavior.

---

> ### Author Rebuttal · Authors · 2026-03-30
>
> We thank the reviewer for recognizing the **real gap** addressed by our work, the **coherent pipeline**, and **visually appealing results**.
>
> All referenced tables, figures, and animated results are available at: https://anon-supp.github.io/
>
> ## On the novelty of our approach
>
> Our **core contribution** is in the structured integration of techniques enabling **360° immersive and navigable text-to-3D world generation**, a capability not achieved by prior work, which inherently trade off immersivity for navigability, or vice versa. To our knowledge, our method is the **first** to jointly achieve both.
>
> Key novel ideas include: (1) using **spherical building blocks** from panoramas, designed for omnidirectional coverage while remaining composable; (2) a **panorama-fusion pipeline** connecting adjacent spheres via midpoint rendering, inpainting, and 3D lifting for consistent expansion. The novelty lies in **how these components are unified** into a scalable generative pipeline.
>
> ## On the importance of Harmonic Blending
>
> While harmonic blending builds on classical formulations, our contribution is in **adapting it to fuse newly generated depth with existing geometry** in an iterative text-to-3D pipeline, ensuring geometric continuity during expansion.
>
> To support our design choice, we provide **new experiments**: (1) an ablation replacing harmonic blending with three alternatives within our pipeline, and (2) an in-depth analysis on the **task of depth completion**.
>
> ### Ablation within our pipeline
>
> We conduct **new experiments** where we replace Harmonic Blending (HB) with naïve blending, bilinear interpolation, and InFusion [2] (diffusion-based depth inpainting).
>
> **Qualitatively** (Section B, Figure 2), HB outperforms all alternatives, fully completing all missing regions.
>
> **Quantitatively**, we compute C-CLIP (semantic consistency), CLIP-Score (text alignment), BRISQUE, CLIP-IQA, and Q-Align (rendering quality). HB outperforms all alternatives on every metric (Section B, Table 2).
>
> ### In-depth analysis
>
> We conduct **new experiments** where we mask portions of reference depth maps and reconstruct them with each method.
>
> **Qualitatively** (Section E, Figure 5), HB provides the smoothest transitions without artifacts.
>
> **Quantitatively** (Section E, Table 4), we report: **Depth Transition Score** (avg. absolute depth difference across the known/reconstructed boundary; lower = smoother) and **Depth Estimation Error** (MAE vs. ground-truth in a narrow band near the boundary). Our method achieves the best scores on both metrics.
>
> ## World consistency
>
>
> We thank the reviewer for raising this important point.
>
> In preliminary experiments, we explored cross-view conditioning strategies and observed **progressive semantic drift**, consistent with prior iterative completion methods [1]. As the scene expands, conditioning becomes increasingly biased toward recent views, leading to a loss of consistency with earlier content.
>
> To address this, we adopt a **shared implicit memory strategy** by using the input prompt as a global conditioning signal across all panorama generations. We observe that modern text-to-image models maintain **strong semantic consistency across independently generated views** when the prompt is sufficiently detailed, effectively acting as a shared scene prior.
>
> A key advantage of this design is that it avoids error accumulation over long trajectories, ensuring that distant regions remain as semantically consistent as nearby ones.
>
> Combined with our geometric alignment (via fusion and blending), this approach preserves consistency without requiring explicit global optimization
>
> We further support this with **new quantitative evaluations** using CLIP-Score and C-CLIP, which measure alignment with the prompt and cross-view consistency, respectively. Our method achieves the best performance under combined rotation and translation, indicating strong global consistency (Section G, Table 5).
>
> We will clarify this design choice in the final version of the paper.
>
> ## Bigger worlds
>
> We provide **new experiments** with larger worlds.
>
> **Qualitatively** (Section A, Figure 1), SphericalDreamer produces long-range environments.
>
> **Quantitatively** (Section A, Table 1), metrics (BRISQUE, Coverage, CLIP-Score, C-CLIP, CLIP-IQA, Q-Align) remain stable as N increases from 3 to 7, confirming maintained quality and consistency.
>
> ---
>
> We sincerely hope that our response resolves the concerns raised. If the reviewer has other technical points to discuss that may currently prevent them from reconsidering the score, we would be more than happy to discuss further.
>
> ## References
>
> [1] Chung, J. et al. (2025). LucidDreamer: Domain-Free Generation of 3D Gaussian Splatting Scenes. IEEE TVCG, 31(12), 10640–10651.
>
> [2] Liu, Z. et al. (2024). InFusion: Inpainting 3D Gaussians via Learning Depth Completion from Diffusion Prior. arXiv:2404.11613.

---

> > ### Author Rebuttal · Reviewer_dnTD · 2026-04-03
> >
> > Thanks for the detailed rebuttal and the additional experiments. I appreciate the effort and agree that this is a challenging and important problem.
> >
> > However, my main concern still remains. The central issue is persistent semantic and geometric consistency across multiple panoramas. In this pipeline, errors can accumulate through projection, panorama linking, inpainting, and depth estimation. I am not convinced that object identity, structure, and semantics remain stable over longer navigation paths.
> >
> > The added CLIP-based metrics are helpful, but they are still indirect proxies. In the video demos, some structures that should remain stable, such as straight trees, still appear inconsistent across views. So although the results are immersive, I am not yet convinced that the method achieves true global consistency.
> >
> > For this reason, I will keep my score unchanged.

---

> > > ### Author Response · Authors · 2026-04-03
> > >
> > > The reviewer reiterates their concern that semantic and geometric errors may accumulate along long navigation paths. We respectfully clarify that this **does not apply** to our setting.
> > >
> > > Hypothetically, such errors could arise from two sources:
> > > - (i) **Model-induced errors.** A high variability in the generated panoramas, in the case where the generative model exhibits significant variance, would break cross-panorama consistency.
> > > - (ii) **Accumulative errors.** Errors that accumulate across successive generations, in the case where panorama generations depend on previously generated outputs, would degrade world consistency as the world size grows.
> > >
> > > We show below that neither condition holds for SphericalDreamer, and therefore **our method is not subject to accumulative semantic or geometric errors**.
> > >
> > > ## (i) Low variability in the generated panoramas
> > >
> > > Prior work show that, for text-conditioned image generative models, as the conditioning prompt becomes more specific, the diversity of images produced by diffusion models drops sharply. [1,2] demonstrate that increasing prompt complexity leads to lower conditional diversity, and that diversity declines and then plateaus as the prompt length grows.
> > >
> > > In other words, very detailed prompts narrow the model’s conditional distribution so much that different random seeds produce outputs with consistent colors, layout, object identity and overall semantics.
> > >
> > > To further support this claim, we updated our [anonymous website](https://anon-supp.github.io/) with a new experiment (Section I, Figure 6) showing multiple panoramas generated by the same model under an identical prompt, showing that the resulting panoramas remain semantically consistent.
> > >
> > > Together, these arguments support that the generated 3D worlds **remain consistent along long paths**, in terms of semantics, size, layout, style and colors, with the text prompt effectively acting as a shared world memory.
> > >
> > > ## (ii) No error accumulation
> > >
> > > In our framework, panoramas are sampled independently from the same prompt-conditioned generative model (i.i.d.). Under this assumption, there is no mechanism by which errors can compound over time. To rigorously address this point, we provide both:
> > > - (ii.a) a theoretical analysis, and
> > > - (ii.b) empirical evidence,
> > >
> > > demonstrating that **errors do not accumulate with increasing path length** in our setup.
> > >
> > > ### (ii.a) Theoretical justification
> > >
> > > Let $I_0, I_1, I_2, \dots \overset{\text{i.i.d.}}{\sim} P(\cdot \mid \text{prompt})$ be panoramas sampled independently from the same prompt-conditioned distribution P.
> > >
> > > Let $Z_n := \mathrm{CLIP}(I_n) \in \mathbb{R}^d$ denote their CLIP embeddings.
> > >
> > > Since CLIP is deterministic, the $Z_i$ are also i.i.d. In particular, for any $n>0$, $Z_n$ and $Z_0$ are independent and identically distributed. Therefore,
> > >
> > > $\mathbb{E}\big[\langle Z_n, Z_0\rangle\big]=\mathbb{E}[Z_n]^\top \mathbb{E}[Z_0]=\|\mathbb{E}[Z]\|^2,$
> > >
> > > which does not depend on n.
> > >
> > > This indicates that the expected semantic similarity between $I_n$ and $I_0$ is **invariant with respect to $n$**. Consequently, there is **no accumulation of semantic error with path length**.
> > >
> > > ### (ii.b) Empirical justification
> > >
> > > In the additional experiments provided in the rebuttal, we generate worlds of varying sizes $N$ and evaluate their properties quantitatively. These results confirm that our generated worlds maintain consistent levels of quality, global semantic consistency, and prompt alignment as $N$ increases.
> > >
> > > Specifically, we report:
> > > - Consistency: CLIP-score, C-CLIP
> > > - Quality: BRISQUE, CLIP-IQA, Q-Align
> > >
> > > As shown in the table below, all metrics remain stable across increasing values of N.
> > >
> > > This provides direct **empirical evidence that errors do not accumulate with path length**, but instead remain stable, in line with our theoretical analysis.
> > >
> > > | Setting | BRISQUE (↓) | Coverage (↑) | CLIP-score (↑) | C-CLIP (↑) | CLIP-IQA (↑) | Q-Align (↑) |
> > > |:-------|:---:|:---:|:---:|:---:|:---:|:---:|
> > > | N=3 | 43.4683 | 0.9993 | 0.3418 | 0.8608 | 0.7582 | 2.3662 |
> > > | N=4 | 42.2766 | 0.9993 | 0.3418 | 0.8471 | 0.7387 | 2.3057 |
> > > | N=5 | 41.548 | 0.9994 | 0.3403 | 0.8447 | 0.7747 | 2.3088 |
> > > | N=6 | 42.1268 | 0.9995 | 0.34 | 0.8499 | 0.7777 | 2.312 |
> > > | N=7 | 41.5514 | 0.9994 | 0.3424 | 0.8453 | 0.7913 | 2.349 |
> > >
> > > ## Conclusion
> > >
> > > In conclusion, the combination of these results ensures that our method does not suffer from consistency issues. Point (i) shows that independently sampled panoramas are semantically consistent, while point (ii) demonstrates that this property holds regardless of the size of the generated world.
> > >
> > > In light of these clarifications, we respectfully invite the reviewer to reconsider their concern regarding the accumulation of semantic and geometric errors.
> > >
> > > ## References
> > >
> > > [1] Zhang et al., The Intricate Dance of Prompt Complexity, Quality, Diversity and Consistency in T2I Models, ICLR 2026.
> > >
> > > [2] Jin et al., Stage-wise Dynamics of Classifier-Free Guidance in Diffusion Models, ICLR 2026.

---

### Decision · Program_Chairs · 2026-04-30

**Decision:**

Accept (regular)

**Comment:**

The paper addresses an interesting text-to-3D world generation problem by aiming to support both full immersive viewing and longer-range navigability. Reviewers generally found the paper clear, technically solid, and supported by strong qualitative results.

The initial reviews raised concerns about limited evaluation, cross-view/global consistency, and the breadth of validation.
Three reviewers found the rebuttal fully resolved their concerns. One reviewer remains doubtful about whether the method can achieve true global consistency. The authors provided additional results to show that their method does not suffer from consistency issues.
Overall, AC finds that all the additional tests and the positive reviews are enough to recommend acceptance.